# Truthful Aggregation of LLMs with an Application to Online Advertising

## Abstract

The next frontier of online advertising is revenue generation from LLM-generated content. We consider a setting where advertisers aim to influence the responses of an LLM to align with their interests, while platforms seek to maximize advertiser value and ensure user satisfaction. The challenge is that advertisers' preferences generally conflict with those of the user, and advertisers may misreport their preferences. To address this, we introduce MOSAIC, an auction mechanism that ensures that truthful reporting is a dominant strategy for advertisers and that aligns the utility of each advertiser with their contribution to social welfare. Importantly, the mechanism operates without LLM fine-tuning or access to model weights and provably converges to the output of the optimally fine-tuned LLM as computational resources increase. Additionally, it can incorporate contextual information about advertisers, which significantly improves social welfare. Through experiments with a publicly available LLM, we show that MOSAIC leads to high advertiser value and platform revenue with low computational overhead. While our motivating application is online advertising, our mechanism can be applied in any setting with monetary transfers, making it a general-purpose solution for truthfully aggregating the preferences of self-interested agents over LLM-generated replies.

## 1 Introduction

*Large language models (LLMs)* are becoming ubiquitous – as coding assistants, as chat interfaces, and even as alternatives to search engines (Bommasani et al., 2022). To ensure their usefulness, it is essential to closely align LLM outputs with user preferences. However, in general, there may be multiple interested parties who disagree on the desired behavior of *the same* LLM. This leads to the question of *how to generate LLM replies given multiple conflicting preferences*.

The motivating application for our work is online advertising, which is the main source of revenue for large tech companies such as Google, Meta, and Twitter. For example, in 2023, Meta's advertising revenue of 132 billion USD was more than 97% of its total revenue (Meta, 2024). Ad auctions are the workhorse mechanism to determine the placement and prices of commercial content (Varian, 2007; Edelman et al., 2007). As platforms are now working on integrating advertisements into LLM-generated content (Criddle, 2024), new auction mechanisms are needed for this application.

We present a new auction mechanism for this problem, which we call *MOSAIC (**M**echanism for **O**ptimally **S**ampling and **A**ggregating LLM Outputs with **I**ncentive **C**ompatibility)*. The advertisers are paying, not for some particular item or bundle, but rather to influence the output generated by an LLM in a direction closer to their own preferences. Although our mechanism could be of interest whenever one has to aggregate the preferences of multiple self-interested agents over LLM behavior (as long as it is reasonable to charge monetary payments), we see online advertising as the most natural setting of interest. For this reason, we refer to participants in the mechanism as *advertisers*.

### 1.1 Problem setting

Consider a situation where a user queries an LLM with a specific question. We assume the following:

- There is a *reference LLM* that can produce useful replies to the user's query.
- There are *advertisers*, each with their own LLM, who want to influence the reply to the user.

In our model, each advertiser is represented via her own LLM or directly with a *reward function*, similar to the function used to fine-tune LLMs in the *Reinforcement Learning from Human Feedback* pipeline (Azar et al., 2023). For this reason, we refer to an advertiser's value for a reply as her *reward*. The auctioneer's task is, given the user's query and the advertisers' preferences, to *produce a reply that is useful for the user while also generating high rewards for the advertisers*.

**Example.**    A user queries an LLM *"What are effective ways to learn a musical instrument online?"* There are two advertisers: *MusicMastery*, which offers online music lessons, and *InstaTune*, which sells musical instruments. MOSAIC merges the preferred replies from the advertisers' LLMs and the reference LLM to produce one combined reply to the user:

> *Effective ways to learn a musical instrument online include:*
> *1. Online music lessons: Platforms like* **MusicMastery** *offer one-on-one virtual tutoring, group classes, and pre-recorded lessons.*
> *2. Interactive learning tools:* **InstaTune** *offers a range of interactive learning tools, such as apps, software, and games, that make learning fun and engaging.*

### 1.2 OVERVIEW OF CONTRIBUTIONS

In this paper, we present MOSAIC, an auction mechanism designed to aggregate the preferences of multiple self-interested advertisers over LLM-generated replies. The first challenge we address is that MOSAIC must produce replies for which the advertisers receive high rewards, but without steering the LLM's behavior too far from that of the user-centric reference LLM. We address this by drawing a strong connection to the RLHF objective (e.g., Ziegler et al. (2020)), equipping MOSAIC with a hyperparameter that enables the auctioneer to balance between the usefulness of the produced reply to the user and the advertisers in an interpretable and principled way (Section 3.1).

Second, leveraging an importance sampling-based technique, MOSAIC is the only mechanism in the literature that converges to the optimal distribution as computational resources increase (Corollary 4.1). Third, MOSAIC can incorporate contextual information, similar to how sponsored search auctions utilize advertiser descriptions. This accelerates convergence and increases value for the advertisers and revenue for the auctioneer (Section 6.2.1).

Fourth, we address technical feasibility and practicality. We adopt the desiderata proposed by Duetting et al. (2023), who argued that auctions must be compatible with existing LLM technology, only using "information obtainable from current models" in such a way that they are "easy to integrate into the system" and rely only on "easy manipulations of LLM output." Furthermore, it must be computationally feasible to run the auction repeatedly and with different advertiser preferences. In particular, we cannot afford the expensive process of fine-tuning an LLM's weights for each possible query. To address this, we have designed MOSAIC to work by only post-processing multiple LLM outputs – it requires only "API access", not even viewing the model weights ( Section 4.2).

Fifth, we address the fact that advertisers might have an interest in misreporting their preferences (which include their *full LLMs*), analogous to over-/underbidding in traditional auctions. To address this challenge, we employ an allocation rule that is compatible with Rochet payments (Rochet, 1987). These payments uniquely ensure that MOSAIC is strategyproof (i.e., that truthful reporting is a dominant strategy for each advertiser, no matter what the others do (Theorem 5.1)). Crucially, Rochet payments ensure that MOSAIC is strategyproof even if the allocation rule has not converged.

Sixth, we introduce an offset to the Rochet payments, to ensure that each advertiser's utility is proportional to her contribution to the social welfare. This alignment is important for the long-term success of the market, incentivizing only the most relevant advertisers to participate.

Finally, we discuss *individual rationality (IR),* which guarantees that a participant is weakly better off from participating in the mechanism. In Section 5.2, we discuss the unique properties of our setting that make *ex-post* IR unattainable. However, in Section 5.3.1, we show that MOSAIC is "almost ex-post IR" in a well-defined sense. In Section 6.2.1, we experimentally show that, for the distribution of instances tested, it is *ex-ante* IR (i.e., advertisers are in expectation better off by participating).

In Section 6, we provide experimental results for the online advertising domain. We demonstrate that MOSAIC quickly converges to the optimal LLM with low computational cost, generating significant value for the advertisers and revenue for the auctioneer while also being useful to the user.

## 2 RELATED WORK

Duetting et al. (2023) introduced the field of mechanism design for LLMs. They proposed a sequential mechanism, where the output sequence is generated on a token-by-token basis and the advertisers bid for their LLM to generate the next token. However, their approach suffers from several limitations: (i) Advertisers face the *exposure problem*: small changes in the generated sequence, such as the addition of a word like "not," can completely alter its meaning. An advertiser might pay a significant amount for the tokens generated up to a certain point, only for the continuation to unexpectedly negate or distort her intended message.[1] (ii) The mechanism is easily manipulable if the assumption that advertisers cannot misreport their LLMs is dropped. (iii) For a given prompt, an advertiser's spend grows with the length of the generated sequence. MOSAIC does not have any of these limitations.

Dubey et al. (2024) proposed a generalization of the position auction (Varian, 2007; Edelman et al., 2007) to a setting where each advertiser is interested in having some specific text ad displayed alongside the organic results. In their mechanism, an LLM module coupled with an auction module work in tandem to merge the ads into a single ad summary in an incentive-compatible way.

Hajiaghayi et al. (2024) considered advertisers bidding in an auction to have their ads placed into various portions of a retrieval-augmented generation (RAG) response. Their auction only allows advertisers to report (and misreport) scalar values to influence the inclusion of non-misreportable advertising texts in the combined output. This is in contrast to MOSAIC, which allows the aggregation of full LLMs that advertisers may arbitrarily misreport.

Feizi et al. (2024) presented an abstract design for an LLM advertising system and detailed a number of research challenges that would have to be overcome in the course of implementation. Conitzer et al. (2024) drew connections between *computational social choice* and LLM alignment. Fish et al. (2023) presented work in the opposite direction: how can LLMs be used to solve problems in social choice? Harris et al. (2024) studied Bayesian persuasion in an abstract setting where a "simulator" (for example, a realistic LLM) of the agent is available. Werner et al. (2024) gave preliminary empirical evidence that LLM-generated content can influence consumers.

## 3 FRAMING SEQUENCE GENERATION AS A MECHANISM DESIGN PROBLEM

### 3.1 FORMAL MODEL

A *user* issues a query $x$. There is a *reference LLM* $\pi_{\text{ref}}$ that the auctioneer aims not to deviate from too much (e.g., because it is responsible for providing useful replies to the user). Additionally, there is a set $N$ of $n$ advertisers who have their own preferences for the reply (i.e., a token sequence) that will be returned to the user. We use the terms *sequence* and *reply* interchangeably.

An LLM can be abstracted as a mapping from (partial) token sequences to a distribution over the next token, or equivalently as an implicit distribution over token sequences. We use this second abstraction, i.e., $\pi_i(y|x)$ denotes the probability that advertiser $i$'s LLM $\pi_i$ assigns to output sequence (i.e., reply) $y$ for the user query $x$.

We let $r_i(x, y)$ denote advertiser $i$'s *reward* for sequence $y$, given query $x$.[2] Informally, the auctioneer's goal is to sample the final sequence from a distribution that optimizes the advertisers' expected rewards without substantially diverging from $\pi_{\text{ref}}$. Formally, the goal is to maximize:

$$J(\pi) = \mathbb{E}_{y \sim \pi} \left[ \sum_{i \in N} r_i(x, y) \right] - \tau D_{\text{KL}}(\pi(\cdot|x) || \pi_{\text{ref}}(\cdot|x)) \qquad (1)$$

where $\tau > 0$ is a hyperparameter enabling the auctioneer to balance producing replies closer to the reference LLM or with higher reward for the advertisers, and $D_{\text{KL}}$ is the Kullback-Leibler divergence.

---

[1]As an example, suppose advertiser A has successfully bid for all the tokens in the sequence "Planning your next vacation? For the cheapest flights to Hawaii, book via..." However, she fails to win the bid for the crucial next token, which would have been her brand name. This leaves her having paid for the preceding tokens without getting the benefit of associating her brand with the message.

[2]In theory, the advertiser's reward for a generated sequence could also depend on user-specific information, but we abstract that away. Equivalently, we assume that the reward function $r_i$ provided by advertiser $i$ is specific to the given user that asked the query $x$.

This objective is similar to the standard Reinforcement Learning from Human Feedback (RLHF) approach (Ziegler et al., 2020), but replaces the human feedback reward function $r_{\text{HF}}(x, y)$ with the sum of the advertisers' rewards. For an overview of RLHF, we recommend Rafailov et al. (2023, §3).

The optimal solution to the optimization problem in (1) was derived by Peters & Schaal (2007):

$$\pi_r(y|x) = \frac{1}{Z(x)} \pi_{\text{ref}}(y|x) \exp\left(\frac{1}{\tau} \sum_{i \in N} r_i(x, y)\right), \tag{2}$$

where $Z(x) = \sum_{y \in T^*} \pi_{\text{ref}}(y|x) \exp\left(\frac{1}{\tau} \sum_{i \in N} r_i(x, y)\right)$ is the partition function.

Let $\vec{R}$ be the set of all possible reports by the advertisers. A *mechanism* is defined as a pair $(\pi, p)$. The *allocation rule* $\pi : \vec{R} \to \Delta(T^*)$ maps any report profile $\vec{r} = (r_1, r_2, \ldots, r_n) \in \vec{R}$ of the advertisers' rewards to a distribution over sequences $\delta(T^*)$. We denote the advertisers' *aggregate reward* as $r(x, y) = \sum_{i=1}^{n} r_i(x, y)$, and their reward profile as $\vec{r}(x, y) = (r_1(x, y), r_2(x, y), \ldots, r_n(x, y))$. The *payment rule* $p : \vec{R} \to \mathbb{R}^n$ maps any report profile of the advertisers' rewards to a payment profile $\vec{p}$, where $\vec{p}_i$ is the payment of the $i$-th advertiser to the mechanism.

A mechanism is *strategyproof* if and only if, for any advertiser $i \in N$, truthfully reporting her rewards is always the optimal strategy, regardless of the reports of other advertisers. In our setting, this holds in expectation because the reply is sampled from the optimal LLM for the platform's objective, based on the advertisers' reports. Formally:

**Definition 3.1** (Strategyproof). *A mechanism $(\pi, p)$ is* strategyproof *if, for all advertisers $i \in N$, for all true reward functions $\vec{r}_i \in \vec{R}_i$, for all reports $\vec{r}_{-i} \in \vec{R}_{-i}$ by the other advertisers, and for all possible misreports $\vec{r}'_i \in \vec{R}_i$ by advertiser $i$: $\mathbb{E}_{y \sim \pi(\vec{r})}[u_i(y, \vec{r}_i, \vec{r}_{-i}; r_i, x)] \geq \mathbb{E}_{y' \sim \pi(\vec{r}'_i, r_{-i}, x)}[u_i(y, \vec{r}'_i, \vec{r}_{-i}; \vec{r}_i)]$, where $u_i(y, \vec{r}_i, \vec{r}_{-i}; r_i, x) = r_i(x, y) - p_i(\vec{r})$ is the utility of advertiser $i$ for reply $y$ when her payment is $p_i(\vec{r})$.*

### 3.2 THE SHORTCOMINGS OF VCG IN THIS SETTING

At first sight, it may seem that the Vickrey–Clarke–Groves (VCG) mechanism would be suitable for our setting (Vickrey, 1961; Clarke, 1971; Groves, 1973). VCG selects the outcome that maximizes the sum of all agents' values. This can be either the single optimal sequence in Equation (1) or the optimal distribution of Equation (2). The VCG mechanism has a corresponding payment rule to incentivize truthful reporting: it charges each agent her externality, that is, the total reduction in value (respectively expected value) for the other agents that her participation in the mechanism caused.

However, in our setting, VCG is not a viable option: The VCG allocation rule requires calculating the *exact* optimal solution to the optimization problem, which is intractable for choosing an LLM to maximize Equation (1) and is even difficult for choosing a single optimal sequence. If a sub-optimal solution is chosen, VCG's strategyproofness is no longer guaranteed (Nisan & Ronen, 2007; 1999; Lehmann et al., 2002). Thus, VCG is not a suitable mechanism in our setting.

## 4 THE MOSAIC MECHANISM: ALLOCATION RULE

### 4.1 CONVERGENCE TO OPTIMALITY, ADVERTISER CONTEXTS AND IMPORTANCE SAMPLING

In this section, we introduce MOSAIC's allocation rule. The high-level idea is that first, a set of $M$ *candidate replies* is generated based on some LLM $\pi_{\text{gen}}$ responsible for that task. Then, the probability of returning each candidate reply is re-weighted based on the advertisers' reports and the reference LLM $\pi_{\text{ref}}$ so that in the limit, as $M \to \infty$, the probability of returning each reply converges to its probability under the optimal distribution of Equation (2). This approach shares similarities with the rejection sampling approach that has been used at various points in the LLM training pipeline (e.g., Bai et al. (2022)). We defer all proofs to Appendix A.1.

**Corollary 4.1.** *The limiting policy $\pi_{\theta, M}(\cdot|x)$ induced by Algorithm 1 is the KL regularized policy that optimizes the aggregate reward function:*

$$\lim_{M \to \infty} \pi_{\theta, M}(\cdot|x) = \arg\max_\pi \mathbb{E}_{y \sim \pi(\cdot|x)}[r(x, y)] - \tau D_{KL}(\pi || \pi_{ref}) \tag{3}$$

---

**Algorithm 1:** Allocation Rule for MOSAIC

---

**Input:** User prompt $x$, reference LLM $\pi_{\text{ref}}$, LLM used for candidate reply generation $\pi_{\text{gen}}$, advertiser reward functions $\vec{r}$, number of candidate replies to generate $M$, reference LLM weight $\tau$

**Output:** Reply $y$ drawn according to the optimal distribution as defined in Equation (1) for the aggregate reward function $r(x, y) = \sum_{i=1}^{N} r_i(x, y)$

**1** Sample $y_j \sim \pi_{\text{gen}}(\cdot|x)$, $1 \leq j \leq M$

**2** Calculate $r(x, y_j) = \sum_{i=1}^{N} r_i(x, y_j)$, $1 \leq j \leq M$

**3** **return** $y \sim \text{softmax}\left(\frac{r(x,y_1)}{\tau} + \log \frac{\pi_{\text{ref}}(y_1|x)}{\pi_{\text{gen}}(y_1|x)}, \ldots, \frac{r(x,y_M)}{\tau} + \log \frac{\pi_{\text{ref}}(y_M|x)}{\pi_{\text{gen}}(y_M|x)}\right)$

---

Based on Corollary 4.1, MOSAIC converges to the optimal distribution for *any* LLM $\pi_{\text{gen}}$ such that $\pi_{\text{ref}}$ is absolutely continuous with respect to $\pi_{\text{gen}}$. The intuitive choice for $\pi_{\text{gen}}$ is $\pi_{\text{ref}}$. However, the practical convergence rate would be very slow. The reason is that $\pi_{\text{ref}}$ may only generate replies with high rewards for the advertisers exceedingly rarely. Advertisers have high rewards for responses that explicitly mention their brands, but $\pi_{\text{ref}}$ considers replies with mentions of specific brands extremely unlikely. Thus, if we used $\pi_{\text{ref}}$ to generate the candidate replies, Algorithm 1 would sample the final reply from a set of candidate replies with low rewards for the advertisers.

To address this challenge, we generate candidate sequences not from $\pi_{\text{ref}}(\cdot|x)$, but instead from a *context-aware LLM*, $\pi_{\text{con}}(\cdot|x; \vec{c})$. The instance-specific context $\vec{c}$ is designed to bridge the gap between the parts of the output space favored by $\pi_{\text{ref}}$ and those valued by the advertisers.

In Appendix A.2, we formally motivate the usage of $\pi_{\text{con}}(\cdot|x; \vec{c})$. Our allocation rule can be interpreted as an importance-based sampling estimator that simulates the probability with respect to the optimal LLM $\pi_r$ of a single sample drawn from $\pi_r$. We show that the estimator is unbiased and that its variance is $\text{Var}(\widehat{\mu}_{\text{IS}}|x) = \frac{1}{M}\left(\sum_{y \in Y} \frac{\pi_r(y|x)^4}{\pi_{\text{gen}}(y|x)} - \left(\sum_{y \in Y} \pi_r(y|x)^2\right)^2\right)$. The estimator's variance is inversely proportional to the number of candidate replies and proportional to $\frac{\pi_r(y|x)^4}{\pi_{\text{gen}}(y|x)}$. Thus, generating the replies via an LLM $\pi_{\text{con}}(\cdot|x; \vec{c})$ that more closely matches $\pi_r(\cdot|x)$ reduces the estimator's variance.

In our application of integrating advertisers' interests into LLM outputs, $\vec{c}_i$ is a context-specific description of the $i$-th advertiser. These descriptions, supplied by the advertisers themselves, should be easily verifiable and factually accurate, akin to "MusicMastery: offering online music lessons", or "InstaTune: selling musical instruments."[3] This approach is analogous to search engine optimization in sponsored search advertising, where advertisers supply and potentially optimize their own descriptions to influence how they are presented by the auction mechanism.

Our experiments in Section 6.2.1 demonstrate that, when using reasonable computational resources, using the context-aware LLM to generate candidate replies achieves substantially higher rewards and utility for the advertisers, increased revenue for the auctioneer, and faster convergence. In the rest of the paper, we refer to using the reference and context-aware LLMs as the baseline and context-aware versions of our mechanisms, respectively.

## 4.2 Applicability and Practical Considerations of MOSAIC

**Generation of Candidate Sequences.** The computationally most expensive part of our mechanism is the generation of candidate sequences, which requires an LLM inference call for the generation of each candidate sequence and potentially $n$ additional calls to calculate all advertisers' rewards. Thus, the computational cost of our mechanism scales linearly in the number of candidate sequences, underscoring the importance of approximating the optimal distribution with few candidate sequences.

**Input Methods and Computational Efficiency.** The mechanism's allocation and payment rules do not depend on the advertisers' full reward functions, but only on their rewards for the candidate

---

[3] A practical way of implementing $\pi_{\text{con}}(\cdot|x; c)$ given $\pi_{\text{ref}}(\cdot|x)$ is to augment the input $x$ to the reference LLM with the advertiser descriptions. In our example: "Try to mention ⟨advertiser x⟩, ⟨advertiser x description⟩.'

sequences. Thus, our mechanism requires only "API access" to the involved LLMs without fine-tuning or access to their weights. Rafailov et al. (2023) established a mapping between an agent's LLM and her implicit reward function, allowing our mechanism to use as inputs sequence probabilities (i.e., LLM inference calls) instead of rewards. For more details, see Appendix B.2.

**Reward elicitation.** Our mechanism can be integrated with computationally efficient methods for eliciting advertisers' reward functions and sampling from $\pi_{con}$, reducing computational overhead. For instance, Li et al. (2024) demonstrated that a simple linear function on a model's embedding space can approximate an LLM's reward function. If advertisers' reward functions are represented this way, an auction instance with $M$ candidate sequences and $n$ advertisers would require only $M$ LLM inference calls instead of $n \cdot M$, and $n \cdot M$ linear multiplications, reducing overhead by a factor of $n$.

**Parallelization.** The generation and evaluation of each candidate sequence are independent processes. This independence allows our mechanism to be fully parallelized, ensuring that the response time for a user query is comparable to that of directly querying a single LLM.

**Inherent Competition.** In conventional auctions, competition among bidders is crucial for driving up prices; when the market is thin, prices tend to be low. Auctioneers often use reserve prices to mitigate this problem. However, reserve prices are challenging to calibrate and can lead to lost revenue if not optimized properly. In our design, advertisers inherently compete with the reference LLM, ensuring consistent competition even in thin markets. This built-in competition mitigates the need for reserve prices and ensures a baseline revenue even with only one advertiser.

## 5 THE MOSAIC MECHANISM: PAYMENT RULE

In this section, we first show how the allocation rule from Section 4 can be combined with an appropriate payment rule so that the resulting mechanism is strategyproof (Section 5.1). Then, we detail how auctions for LLM-generated content differ from standard auctions (Section 5.2). Taking those differences into account, we create a payment offset, so the mechanism is both strategyproof and social welfare aligned (Section 5.3). We defer all proofs to Appendix B.

### 5.1 STRATEGYPROOF PAYMENTS THROUGH CYCLIC MONOTONICITY

The allocation rule we introduced in Section 4 satisfies *cyclic monotonicity* (Rockafellar, 1970, § 24). Rochet (1987) first proposed the use of cyclic monotonicity in mechanism design as a generalization of the monotonicity property in single-parameter settings (Myerson, 1981). For general settings, cyclic monotonicity of the allocation rule is a sufficient and necessary condition for the existence of a payment rule so that the resulting mechanism is strategyproof. By applying this theory we can prove:

**Theorem 5.1.** *Let $\vec{r}_{-i}$ be the reward profile of all advertisers other than $i$, and let $\vec{r}_{-i}$ be the aggregate reports of all other advertisers. The allocation rule induced by Algorithm 1 can be combined with a payment rule $p : \vec{R} \rightarrow \mathbb{R}^n$ such that in the mechanism $(\pi, p)$ for any advertiser $i \in N$, report profile $\vec{r}_{-i}$ and set of generated candidate replies, reporting truthfully maximizes advertiser $i$'s expected utility, with the expectation taken over the draw of the final reply from the set of candidate replies. Advertiser $i$'s expected utility in $(\pi, p)$ under truthful bidding is:*

$$U_{C,i}(\vec{r}_i, \vec{r}_{-i}; \pi_{ref}, \pi_{gen}) = \tau \log \left( \sum_{j=1}^{M} \exp \left( \frac{1}{\tau} \sum_{k \in N} r_k(x, y_j) \right) + \log \frac{\pi_{ref}(y_j|x)}{\pi_{gen}(y_j|x)} \right) + C, \ C \in \mathbb{R} \tag{4}$$

Note that, based on Theorem 5.1, in our mechanism it is *always* a dominant strategy for an advertiser to report truthfully. Crucially, this is not the case for VCG, where truthful reporting would be optimal only if the allocation rule had converged to the optimal distribution as defined in Equation (2).

### 5.2 DIFFERENCES FROM AUCTION MECHANISMS FOR ITEM ALLOCATION

Auction mechanisms designed to sell items or ad slots typically rely on simplifying assumptions that do not apply in a setting with LLM-generated content. Key differences include: (i) *Non-Negative*

*Values:* These mechanisms assume agents' values are non-negative due to having zero value for the empty bundle and free disposal. In contrast, in our setting, an advertiser's reward can be negative based on the discrepancy between her LLM and the reference LLM. (ii) *Advertiser-Specific Allocations:* These mechanisms allocate different item bundles to different agents. Here, a single reply is produced, and advertisers' rewards depend on that reply's probability with respect to their LLMs. (iii) *Zero Utility for Non-Participation:* In most auction settings, not participating yields zero utility. Here, non-participation can result in negative utility since the produced reply may be unfavorable to non-participating advertisers. For details, see Appendix B.2.

### 5.3 ADVERTISER-SPECIFIC UTILITY OFFSET

We now modify MOSAIC's payment rule by adding a *payment offset* (and thus a utility offset): $C = -U_i(\vec{0}, \vec{r}_{-i}; \pi_{ref}, \pi_{con})$. Informally, we additionally charge each advertiser her utility in Equation (4) for bidding zero for all candidate replies. This offset maintains the nice properties of our mechanism (i.e., strategyproofness and convergence to the optimal distribution), while also achieving two additional properties that are critical for the long-term success of a market for LLM aggregation. The first is *"Almost IR:"* An advertiser with weakly positive reward for all candidate replies has weakly positive expected utility for all reports by the other advertisers. In particular, an advertiser with zero reward for all outputs is guaranteed zero utility. The second is *"What you give is what you get:"* an advertiser's ex-interim expected utility is monotone in how well-aligned her exponentiated reward for the replies is with the interim allocation rule if she were to not participate.

#### 5.3.1 OUR MECHANISM IS "ALMOST INDIVIDUALLY RATIONAL"

Individual rationality is important to incentivize agents to participate in the mechanism. In Appendix B.3, we explain why the standard notion of individual rationality (i.e., weakly positive utility from participation in the mechanism) encountered in standard auction settings is impossible to achieve in our setting while converging to the optimal distribution and maintaining strategyproofness. Then, we explain how, with our payment offset, our mechanism is "almost IR:" In Lemma B.1 we prove that the ex-interim utility of an advertiser who has zero reward for all candidate replies and bids truthfully is deterministically zero, i.e., advertisers that do not contribute to the social welfare (but also do not detract from it) have zero utility. Similarly, in Lemma B.2 we prove that if an advertiser's reward for all candidate sequences is (weakly) positive, then her ex-interim utility is (weakly) positive.

**Remark 1.** *In Section 6, we experimentally show that our offset payment rule, coupled with our context-aware allocation rule, results in both high expected rewards and positive expected utility for the advertisers (i.e., ex-ante individual rationality), as well as significant revenue for the auctioneer.*

#### 5.3.2 "WHAT YOU GIVE IS WHAT YOU GET"

Our allocation rule, which is the only one over a finite set of replies that converges to the optimal LLM, is also the (sub)gradient of the utility to ensure truthfulness (Rochet, 1987). Because the allocation rule is the same for all advertisers, their utilities must also be the same, up to advertiser-specific offsets, as indicated by Equation (4). However, not all advertisers contribute equally to social welfare. In Appendix B.4, we explain why implementing the mechanism without carefully designed offsets would lead to free-riding. Specifically, if an advertiser's utility in Equation (4) is positive, even when her relevance to the user query is low, it incentivizes her participation. This dilutes the utility for all participants, discouraging more relevant advertisers from participating. As a result, the expected rewards for advertisers decline, and the generated replies become less useful for users. To avoid free-riding, it is crucial to align each advertiser's utility with her contribution to social welfare.

**Lemma 5.2.** *For the offset $C = -\tau \log \left( \sum_{j=1}^{M} \exp \left( \vec{\beta}_{-i,j} \right) \right)$ advertiser $i$'s ex-interim utility is:*

$$U_i(\vec{r}_i; \vec{\beta}_{-i}) = \tau \log \left( \sum_{j=1}^{M} \exp \left( \frac{r_i(x, y_j)}{\tau} \right) \pi_{int}(y_j | x; \vec{r}_{-i})_j \right) \tag{5}$$

In words, Lemma 5.2 proves that every agent's ex-interim utility is monotone in how well-aligned the interim allocation rule $\pi_{int}$ (i.e., the probability of returning each of the already generated candidate replies) without her and her exponentiated rewards for the candidate replies are.

| **User query** |
| --- |
| What are effective ways to learn a musical instrument online? |
| **Advertisers** |
| - MusicMastery: offering online music lessons and virtual tutoring 
 - InstaTune: selling musical instruments and learning aids |
| **Context-Aware Mechanism Output** |
| Effective ways to learn a musical instrument online include: 
 1. Online music lessons: Platforms like **MusicMastery** offer one-on-one virtual tutoring, group classes, and pre-recorded lessons. These lessons can be tailored to your skill level and learning style. 
 2. Interactive learning tools: **InstaTune** offers a range of interactive learning tools, such as apps, software, and games, that make learning fun and engaging.... |
| **Baseline Mechanism Output** |
| - Practice consistently and regularly: Consistent practice is essential to learning any musical instrument, and it's even more important when learning online. Set aside a specific time each day or week to practice, and make sure to stick to it. 
 - Find a good teacher or mentor: While it's possible to learn some things on your own, working with a good teacher or mentor can help you improve faster and avoid developing bad habits. ... |

Table 1: Comparison of context-aware and baseline mechanism outputs using 10 candidate sequences. Advertisers highlighted in bold, outputs truncated to conserve space.

**Remark 2.** *In Section 6.2.2, we experimentally show that the offset described in Section 5.3 creates a strong positive correlation between an advertiser's contribution to social welfare and her expected utility gain from participation in the mechanism. Moreover, our experiments show that for the distribution of instances tested, MOSAIC satisfies ex-ante IR.*

# 6 EXPERIMENTS

In this section, we experimentally evaluate the performance of our mechanism. We focus on the online advertising domain, the main motivation application for our mechanism.

## 6.1 EXPERIMENT SETUP

We create synthetic instances consisting of user queries (e.g., "How to learn a musical instrument online?") and advertisers (e.g., "MusicMastery, offering online music lessons"). We use Llama-2-7b-chat-hf as the reference LLM (Touvron et al., 2023). Following Duetting et al. (2023), we create the advertisers' LLMs by adding advertising instructions to the reference LLM. The context-aware LLM is created as described in Footnote 3.

Following Rafailov et al. (2023), advertisers' reward functions are defined as $r_i(x, y) = \log \frac{\pi_i(y|x)}{\pi_{\text{ref}}(y|x)}$. For the auctioneer's objective, we set $\tau = 1$ in Equation (1), balancing advertisers' rewards and divergence from the reference LLM.

We use 50 user queries, each with two advertisers, and test each query on 25 different random seeds, resulting in 1,250 instances. Following Li et al. (2024); Rozière et al. (2024), we sample from all LLMs using a temperature of 0.8 and top-p 0.95. For full experimental details, see Appendix C.1.

## 6.2 EXPERIMENTAL RESULTS

### 6.2.1 EVALUATING THE ALLOCATION RULE

In this section, we evaluate the effectiveness of our allocation rule in generating value for advertisers and converging to the optimal distribution. We assess both the baseline and context-aware versions of MOSAIC, i.e., using the reference and context-aware LLMs to generate candidate replies.

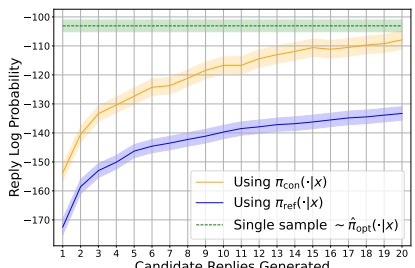
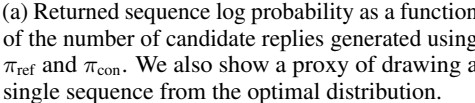
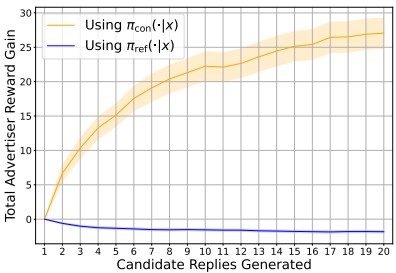

(a) Returned sequence log probability as a function of the number of candidate replies generated using $\pi_{\text{ref}}$ and $\pi_{\text{con}}$. We also show a proxy of drawing a single sequence from the optimal distribution.

(b) Total advertiser normalized reward as a function of the number of candidate replies generated using $\pi_{\text{ref}}$ and $\pi_{\text{con}}$.

Figure 1: Reply log probability and total advertiser normalized reward as a function of the number of candidate sequences generated using $\pi_{\text{ref}}$ and $\pi_{\text{con}}$. Averaged over 1250 runs including 95% CIs.

To provide some intuition on how the context-aware mechanism enhances the relevance of responses for advertisers, we compare outputs from both mechanisms in Table 1. Notably, only the context-aware mechanism successfully incorporates advertisers into the replies.

In Figure 1a, we plot the log probability of the replies returned by MOSAIC with respect to the optimal distribution (Equation (2)) as a function of the number of candidate replies generated. We compare the context-aware version of MOSAIC to the baseline version to evaluate the effectiveness of incorporating contextual information. To benchmark our results, we estimate the log probability of sampling a *single reply* from the optimal distribution.[4]

We observe that for both the context-aware and baseline versions of MOSAIC, the log probability of the returned reply scales rapidly with the number of candidate replies generated. This is in line with our theoretical analysis in Section 4, where we proved that both versions converge to the optimal distribution. We also observe that incorporating context into the mechanism significantly accelerates convergence. Notably, the context-aware mechanism can achieve higher log probability with respect to the optimal distribution with four candidate replies than the baseline version can with 20. Additionally, with only 20 generated candidate replies, the context-aware mechanism can almost match our estimate of the log probability of sampling from the optimal distribution. In Appendix C.3, we further demonstrate that for the context-aware mechanism, the usefulness to the user also scales rapidly with the number of candidate replies.

In Figure 1b, we plot the total expected reward for advertisers as a function of the number of candidate replies generated by the mechanism. The expectation is taken over the draw of the returned reply from the set of candidate replies. To make the plot more interpretable, we normalize an advertiser's expected reward by her expected reward had she not participated in the mechanism. Specifically, we plot $r_{i,\text{norm}}(x) = \sum_{j \in M} r_i(x, y_j) \cdot \pi_{\text{int}}(y_j|x) - \sum_{j \in M} r_i(x, y_{-i,j}) \cdot \pi_{\text{int}}(y_{-i,j}|x)$, where $y_{-i} = (y_{-i,1}, \ldots, y_{-i,M})$ represents the candidate replies that would have been generated had the advertiser not participated.[5] We compare the outcomes for the two versions of our mechanism, highlighting the added value of integrating context.

Figure 1b demonstrates that our context-aware mechanism significantly boosts advertiser rewards, with benefits scaling rapidly as the number of candidate replies increases. In contrast, the baseline version fails to improve advertiser rewards within a computationally feasible number of generated replies, and may even cause a slight decrease. This occurs because generating more replies adds

---

[4]Note that the closed-form solution of Equation (2) allows us to evaluate the probability of sentences with respect to the optimal solution, but it does not enable us to sample from that distribution. Sampling from that distribution would require using reinforcement learning to train the optimal LLM on the advertisers' aggregate reward function, which is computationally infeasible for the number of problem instances we test. Instead, we generate replies from the reference LLM and evaluate them based on the induced probabilities of the reference LLM, for which the reference LLM is the optimal one. This serves as a proxy for the log probabilities we should expect if we were to draw replies from the optimally fine-tuned model for each query.

[5]To reduce computational costs, we estimate an advertiser's reward for not participating based on her expected reward over the already generated replies in which her brand is not mentioned by name, motivated by the fact that if she does not participate, her brand will not be mentioned by name.

marginal positive reward to the reference LLM (see Appendix C.3), but does so at the expense of advertiser rewards. In summary, MOSAIC's support for context-aware LLMs allows it to quickly converge to the theoretically optimal distribution and generate substantial rewards for advertisers.

### 6.2.2 EVALUATING THE PAYMENT RULE

In this section, we examine the impact of our payment rule introduced in Section 5.

In Figure 2, we plot the auctioneer's revenue as a function of the number of candidate replies, for both the context-aware and baseline versions of MOSAIC. We observe that our payment rule in combination with the context-aware allocation rule results in revenue that scales rapidly with the number of generated candidate replies. Furthermore, if we compare the advertisers' total payment in Figure 2 with their normalized reward in Figure 1b, we see that for the context-aware mechanism, our payment rule converts a significant portion of the surplus created for the advertisers into revenue for the auctioneer, while also ensuring positive advertiser utility.

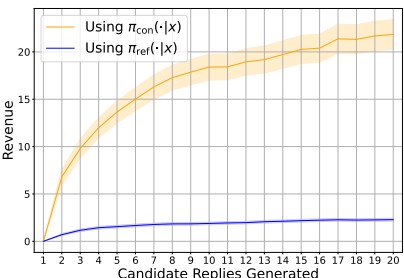

Figure 2: Revenue as a function of the number of replies generated using $\pi_{\mathrm{ref}}$ and $\pi_{\mathrm{con}}$.

These results align with our theoretical analysis. In Section 5.3.1, we showed that, with our payment offset, an advertiser with positive rewards for candidate replies—as in the context-aware mechanism—will achieve positive expected utility. In Section 5.3.2 we additionally introduced the "what you give is what you get" property. In Appendix C.4, we provide a more comprehensive comparison of the relationship between an advertiser's normalized reward and utility, with and without the offset of Section 5.3, for both the context-aware and baseline versions of our mechanism. In all cases, the payment offset increases the advertisers' expected utility, and makes the relationship between an advertiser's contribution to social welfare and utility significantly more linear and positively correlated. Concretely, the Pearson correlation between advertiser utility and reward is $0.8$ with offset payments, compared to $0.4$ with standard Rochet payments.

## 7 CONCLUSION

We have introduced MOSAIC, a novel auction mechanism for aggregating preferences over LLM outputs. MOSAIC provably converges to the theoretically optimal distribution and it also facilitates a principled method for balancing participants' expected rewards with the divergence from a reference LLM. Thus, our mechanism is particularly well-suited for online advertising, allowing the integration of advertiser LLMs with a reference LLM responsible for generating user-centric replies.

A well-chosen payment rule removes any incentive to misreport preferences, achieving the central mechanism design goal of strategyproofness. While ex-post IR is incompatible with strategyproofness in our setting, we experimentally show that our mechanism is ex-ante individually rational and "almost individually rational" in a certain sense. Furthermore, it ensures that each agent's utility gain is proportionate to her contribution to social welfare, an essential alignment property in this setting.

Experimentally, we have demonstrated that by incorporating contextual information, MOSAIC's outputs rapidly converge to the optimal distribution, generating significant value for the advertisers while also effectively recapturing a considerable portion of this value as revenue. These findings demonstrate the practical efficacy and potential of our approach in real-world settings.

There are several interesting directions for future work. While our mechanism converges quickly, developing an approach to estimate the number of candidate replies required on a per-query basis would reduce the platform's costs. Another interesting direction is to develop a formal framework for incorporating context into the mechanism, which could further improve the speed of convergence. Finally, beyond advertising, one could apply a similar mechanism design approach to other problems: for example, as web site owners increasingly demand to be paid for having their content ingested by LLMs, one could consider designing a strategyproof procurement (i.e. reverse) auction to decide which sources are included in an LLM summary.

**Reproducibility statement.** The code, which includes the seeds used for all experimental results in this paper, has been made available to the reviewers and will be made publicly available upon acceptance. Proofs are included in the appendix.

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

# A   DETAILS FROM SECTION 4

## A.1   OMITTED PROOFS FROM SECTION 4

In this section, we present all omitted theorems and proofs from Section 4.

**Theorem A.1.** *Let $\pi_{\theta,M}(y|x)$ be the probability of sampling output sequence $y$ for input sequence $x$ according to Algorithm 1, where $\theta$ is the vector of all input parameters and $M$ is the number of candidate sequences generated. Given the advertisers' reports $\vec{r} \in \vec{R}$, the policy induced by the mechanism approaches the following limit:*

$$\lim_{M \to \infty} \pi_{\theta,M}(y|x) = \pi_{ref}(y|x) \frac{\exp(r(x,y)/\tau)}{\mathbb{E}_{y' \sim \pi_{ref}(\cdot|x)}[\exp(r(x,y')/\tau)]} \tag{6}$$

*Theorem A.1 Proof.* Let $\pi_{\theta,M}(y|x, \{y_j\}_{j=1}^M)$ be the probability of returning output sequence $y$ for input sequence $x$ according to Algorithm 1 conditioned on the set of generated candidate sequences being $\{y_j\}_{j=1}^M$. Additionally, let $\pi_{\mathrm{con}}(\{y_j\}_{j=1}^M|x; \vec{c})$ be the probability of the context-aware model $\pi_{\mathrm{con}}$ generating the candidate sequences $\{y_j\}_{j=1}^M$, given the context $\vec{c}$ and the user query $x$.

First, note that we can write the density of $\pi_{\theta,M}$ as follows:

$$\pi_{\theta,M}(y|x) = \sum_{\{y_j\}_{j=1}^M \in Y^M} \pi_{\theta,M}(y|x, \{y_j\}_{j=1}^M) \pi_{\mathrm{con}}(\{y_j\}_{j=1}^M|x; \vec{c})$$

$$= \mathbb{E}_{\{y_j\}_{j=1}^M \sim \pi_{\mathrm{con}}(\cdot|x;\vec{c})} \left[ \pi_{\theta,M}(y|x, \{y_j\}_{j=1}^M) \right]$$

$$= \mathbb{E}_{\{y_j\}_{j=1}^M \sim \pi_{\mathrm{con}}(\cdot|x;\vec{c})} \left[ \sum_j \mathbb{I}\{y_j = y\} \frac{\exp\left(\frac{r(x,y_j)}{\tau} + \log \frac{\pi_{\mathrm{ref}}(y_j|x)}{\pi_{\mathrm{con}}(y_j|x;\vec{c})}\right)}{\sum_{\zeta \in \{y_j\}_{j=1}^M} \exp\left(\frac{r(x,\zeta)}{\tau} + \log \frac{\pi_{\mathrm{ref}}(\zeta|x)}{\pi_{\mathrm{con}}(\zeta|x;\vec{c})}\right)} \right]$$

$$= \mathbb{E}_{\{y_j\}_{j=1}^M \sim \pi_{\mathrm{con}}(\cdot|x;\vec{c})} \left[ \sum_j \mathbb{I}\{y_j = y\} \frac{\frac{\pi_{\mathrm{ref}}(y_j|x)}{\pi_{\mathrm{con}}(y_j|x;\vec{c})} \exp\left(\frac{r(x,y_j)}{\tau}\right)}{\sum_{\zeta \in \{y_j\}_{j=1}^M} \frac{\pi_{\mathrm{ref}}(\zeta|x)}{\pi_{\mathrm{con}}(\zeta|x;\vec{c})} \exp\left(\frac{r(x,\zeta)}{\tau}\right)} \right]$$

$$= \mathbb{E}_{\{y_j\}_{j=1}^M \sim \pi_{\mathrm{con}}(\cdot|x;\vec{c})} \left[ \frac{\sum_j \mathbb{I}\{y_j = y\}}{\sum_{\zeta \in \{y_j\}_{j=1}^M} \frac{\pi_{\mathrm{ref}}(\zeta|x)}{\pi_{\mathrm{con}}(\zeta|x;\vec{c})} \exp\left(\frac{r(x,\zeta)}{\tau}\right)} \right] \frac{\pi_{\mathrm{ref}}(y|x)}{\pi_{\mathrm{con}}(y|x;\vec{c})} \exp\left(\frac{r(x,y)}{\tau}\right)$$

$$= \mathbb{E}_{\{y_j\}_{j=1}^M \sim \pi_{\mathrm{con}}(\cdot|x;\vec{c})} \left[ \frac{\frac{1}{M} \sum_j \mathbb{I}\{y_j = y\}}{\frac{1}{M} \sum_{\zeta \in \{y_j\}_{j=1}^M} \frac{\pi_{\mathrm{ref}}(\zeta|x)}{\pi_{\mathrm{con}}(\zeta|x;\vec{c})} \exp\left(\frac{r(x,\zeta)}{\tau}\right)} \right] \frac{\pi_{\mathrm{ref}}(y|x)}{\pi_{\mathrm{con}}(y|x;\vec{c})} \exp\left(\frac{r(x,y)}{\tau}\right)$$

Taking the limit as $M \to \infty$ and using the Law of Large Numbers (the sequences are i.i.d.):

$$\lim_{M \to \infty} \pi_{\theta,M}(y|x) = \lim_{M \to \infty} \mathbb{E}_{\{y_j\}_{j=1}^M \sim \pi_{\mathrm{con}}(\cdot|x;\vec{c})} \left[ \frac{\pi_{\mathrm{con}}(y|x;\vec{c})}{\mathbb{E}_{\zeta \sim \pi_{\mathrm{con}}(\cdot|x)} \left[ \exp\left(\frac{r(x,\zeta)}{\tau}\right) \frac{\pi_{\mathrm{ref}}(\zeta|x)}{\pi_{\mathrm{con}}(\zeta|x;\vec{c})} \right]} \right] \frac{\pi_{\mathrm{ref}}(y|x)}{\pi_{\mathrm{con}}(y|x;\vec{c})} \exp\left(\frac{r(x,y)}{\tau}\right)$$

$$= \pi_{\mathrm{con}}(y|x;\vec{c}) \frac{1}{\mathbb{E}_{\zeta \sim \pi_{\mathrm{con}}(\cdot|x)} \left[ \exp\left(\frac{r(x,\zeta)}{\tau}\right) \frac{\pi_{\mathrm{ref}}(\zeta|x)}{\pi_{\mathrm{con}}(\zeta|x)} \right]} \frac{\pi_{\mathrm{ref}}(y|x)}{\pi_{\mathrm{con}}(y|x;\vec{c})} \exp\left(\frac{r(x,y)}{\tau}\right)$$

$$= \pi_{\mathrm{ref}}(y|x) \frac{1}{\sum_{\zeta \in Y} \pi_{\mathrm{con}}(\zeta|x) \exp\left(\frac{r(x,\zeta)}{\tau}\right) \frac{\pi_{\mathrm{ref}}(\zeta|x)}{\pi_{\mathrm{con}}(\zeta|x;\vec{c})}} \exp\left(\frac{r(x,y)}{\tau}\right)$$

$$= \pi_{\mathrm{ref}}(y|x) \frac{1}{\sum_{\zeta \in Y} \pi_{\mathrm{ref}}(\zeta|x) \exp\left(\frac{r(x,\zeta)}{\tau}\right)} \exp\left(\frac{r(x,y)}{\tau}\right)$$

$$= \pi_{\mathrm{ref}}(y|x) \frac{1}{\mathbb{E}_{\zeta \sim \pi_{\mathrm{ref}}(\cdot|x)} \left[ \exp(r(x,\zeta)/\tau) \right]} \exp\left(\frac{r(x,y)}{\tau}\right)$$

$\square$

*Corollary 4.1 Proof.* The proof follows directly from Theorem A.1 and Appendix A.1 in Rafailov et al. (2023).

$\square$

## A.2 FORMAL CONNECTION TO IMPORTANCE SAMPLING

Given truthful reports by the advertisers, there is a strong connection between our allocation rule and importance sampling. Our allocation rule is trying to simulate the probability of drawing a single sequence from the optimal LLM for the platform's objective. The LLM that we use to generate the candidate replies (either the reference LLM or the context-aware LLM) can be interpreted as proposal distribution $\pi_{\text{prop}}(\cdot|x)$ used to generate samples. We are interested in how well, with respect to the target probability $\pi_r(\cdot|x)$ a sample $y$ from our estimator using $K$ candidate replies matches a sample drawn from the target distribution $\pi_r(\cdot|x)$. In this section, we will analytically show how the quality of the proposal distribution $\pi_{\text{prop}}$ affects the quality of that estimator.

Let's assume our allocation rule draws $K$ samples $y_1, y_2, \ldots, y_K$ from the proposal distribution $\pi_{\text{prop}}(\cdot|x)$. The importance sampling estimator for the expected value of the function $h(y|x) = \pi_r(y|x)$ under the target distribution $\pi_r(y|x)$ is

$$\widehat{\mu}_{\text{IS}} = \frac{1}{K} \sum_{j=1}^{K} h(y_j|x) \cdot w(x_j) = \frac{1}{K} \sum_{j=1}^{K} h(y_j|x) \cdot \frac{\pi_r(y_j|x)}{\pi_{\text{prop}}(y_j|x)} = \frac{1}{K} \sum_{j=1}^{K} \frac{\pi_r(y_j|x)^2}{\pi_{\text{prop}}(y_j|x)}$$

The expected value of this estimator is:

$$\mathbb{E}[\widehat{\mu}_{\text{IS}}] = \mathbb{E}_{y \sim \pi_{\text{prop}}(\cdot|x)} \left[ \frac{\pi_r(y|x)^2}{\pi_{\text{prop}}(y|x)} \right] = \sum_{y \in Y} \frac{\pi_r(y|x)^2}{\pi_{\text{prop}}(y|x)} \pi_{\text{prop}}(y|x) = \sum_{y \in Y} \pi_r(y|x)^2 \tag{7}$$

Thus, our estimator is unbiased, as expected. The variance of the estimator $\widehat{\mu}_{\text{IS}}$ is given by:

$$\text{Var}(\widehat{\mu}_{\text{IS}}) = \mathbb{E}[\widehat{\mu}_{\text{IS}}^2] - \mathbb{E}[\widehat{\mu}_{\text{IS}}]^2 \tag{8}$$

For the term $\widehat{\mu}_{\text{IS}}^2$ we have:

$$\widehat{\mu}_{\text{IS}}^2 = \left( \frac{1}{K} \sum_{j=1}^{K} \frac{\pi_r(y_j|x)^2}{\pi_{\text{prop}}(y_j|x)} \right) = \frac{1}{K^2} \sum_{j=1}^{K} \sum_{j'=1}^{K} \frac{\pi_r(y_j|x)^2}{\pi_{\text{prop}}(y_j|x)} \cdot \frac{\pi_r(y_{j'}|x)^2}{\pi_{\text{prop}}(y_{j'}|x)}$$

Taking the expectation $\mathbb{E}_{\pi_{\text{prop}}(\cdot|x)}[\cdot]$:

$$\mathbb{E}[\widehat{\mu}_{\text{IS}}^2] = \frac{1}{K^2} \sum_{j=1}^{K} \sum_{j'=1}^{K} \mathbb{E}_{\pi_{\text{prop}}(\cdot|x)} \left[ \frac{\pi_r(y_j|x)^2}{\pi_{\text{prop}}(y_j|x)} \cdot \frac{\pi_r(y_{j'}|x)^2}{\pi_{\text{prop}}(y_{j'}|x)} \right]$$

$$= \frac{1}{K^2} \left( \sum_{j=1}^{K} \mathbb{E}_{\pi_{\text{prop}}(\cdot|x)} \left[ \frac{\pi_r(y_j|x)^4}{\pi_{\text{prop}}(y_j|x)^2} \right] + \sum_{j \neq j'} \mathbb{E}_{\pi_{\text{prop}}(\cdot|x)} \left[ \frac{\pi_r(y_j|x)^2}{\pi_{\text{prop}}(y_j|x)} \right] \right)$$

$$= \frac{1}{K^2} \left( K \sum_{y \in Y} \frac{\pi_r(y|x)^4}{\pi_{\text{prop}}(y|x)} + K(K-1) \left( \sum_{y \in Y} \pi_r(y|x)^2 \right)^2 \right)$$

$$= \frac{1}{K} \sum_{y \in Y} \frac{\pi_r(y|x)^4}{\pi_{\text{prop}}(y|x)} + \frac{K-1}{K} \left( \sum_{y \in Y} \pi_r(y|x)^2 \right)^2 \tag{9}$$

The final expression for the variance of the importance sampling estimator with $K$ samples can be computed by substituting Equations (7) and (9) in Equation (8):

$$\text{Var}(\widehat{\mu}_{\text{IS}}) = \frac{1}{K} \sum_{y \in Y} \frac{\pi_r(y|x)^4}{\pi_{\text{prop}}(y|x)} + \frac{K-1}{K} \left( \sum_{y \in Y} \pi_r(y|x)^2 \right)^2 - \left( \sum_{y \in Y} \pi_r(y|x)^2 \right)^2$$

$$= \frac{1}{K} \left( \sum_{y \in Y} \frac{\pi_r(y|x)^4}{\pi_{\text{prop}}(y|x)} - \left( \sum_{y \in Y} \pi_r(y|x)^2 \right)^2 \right) \tag{10}$$

Equation ([10]) reveals two important details about our estimator. First, the variance of the estimator is inversely proportional to the number of candidate replies generated $K$. Thus, as $K$ increases the variance decreases and our estimator becomes more stable in simulating the target distribution $\pi_r(\cdot|x)$. The closer that the proposal distribution $\pi_{\text{prop}}(\cdot|x)$ is to the target distribution, the smaller the term $\sum_{y \in Y} \frac{\pi_r(y|x)^4}{\pi_{\text{prop}}(y|x)}$ will be, which reduces the variance of our estimator.

# B    Details from Section 5

In this section, we present all omitted details from Section 5.

## B.1    Omitted Proofs from Section 5.1

*Theorem 5.1 Proof.* Let $\vec{r}_i = (r_i(x, y_1), \ldots, r_i(x, y_M))$ be the reward reports of advertiser $i$ for the $M$ generated candidate sequences. Then, for both allocation rules, holding the candidate sequences and the reports of all other advertisers fixed, the ex-interim allocation rule (i.e., the probability of returning each of the $M$ generated candidate sequences) from advertiser $i$'s perspective is:

$$\pi_{\text{int}}(\vec{r}_i; \vec{\beta}_{-i}) = \text{softmax}\left(\frac{\vec{r}_i}{\tau} + \vec{\beta}_{-i}\right), \tag{11}$$

where $\vec{\beta}_{-i,j} = \frac{\sum_{k \in N \setminus \{i\}} r_k(x, y_j)}{\tau} + \log \frac{\pi_{\text{ref}}(y_j | x)}{\pi_{\text{con}}(y_j | x; \vec{c})}$. Importantly $\vec{\beta}_{-i}$ is an $M$-dimensional vector that does not depend on advertiser $i$'s reports.

We would like to equip $\pi_{\text{int}}$ with a payment rule $p(\vec{r}_i; \vec{\beta}_{-i})$ so that the resulting interim mechanism will be strategyproof. This requires that $\pi_{\text{int}}$ have a property known as *cyclic monotonicity*. Equivalently, $\pi_{\text{int}}$ must be the (sub)gradient of advertiser $i$'s utility for bidding truthfully in he mechanism $U(\vec{r}_i; \vec{\beta}_{-i})$, and that utility function must be convex (Frongillo & Kash, 2021; Rochet, 1987; Myerson, 1981).

It is easy to verify that for the function class:

$$U_C(\vec{r}_i; \vec{\beta}_{-i}) = \tau \log \left(\sum_{j=1}^{M} \exp\left(\frac{r_i(x, y_j)}{\tau} + \vec{\beta}_{-i,j}\right)\right) + C, \ C \in \mathbb{R} \tag{12}$$

the allocation rule $\pi_{\text{int}}(\vec{r}_i; \vec{\beta}_{-i})$ is a gradient of $U_C(\vec{r}_i; \vec{\beta}_{-i})$. Additionally, $U_C(\vec{r}_i; \vec{\beta}_{-i})$ is convex in $\vec{r}_i$: The exponential function $e^x$ is (strictly) convex, because its second derivative is positive. The transformation $\frac{r_i(x, y_j)}{\tau} + \vec{\beta}_{-i,j}$ is an affine transformation of $r_i(x, y_j)$, and affine transformations preserve convexity. Finally, it is well-known that the LogSumExp function is convex.

Thus, for any $\vec{\beta}_{-i}$ and for any set of generated candidate sequences, reporting truthfully maximizes advertiser $i$'s expected utility, with the expectation taken over the draw of the final sequence from the set of candidate sequences. Adopting the quasi-linear utility model, advertiser $i$'s payment is:

$$U_C(\vec{r}_i; \vec{\beta}_{-i}) = \pi_{\text{int}}(\vec{r}_i; \vec{\beta}_{-i}) \cdot \vec{r}_i - p(\vec{r}_i; \vec{\beta}_{-i})$$

$$p(\vec{r}_i; \vec{\beta}_{-i}) = \pi_{\text{int}}(\vec{r}_i; \vec{\beta}_{-i}) \cdot \vec{r}_i - U_C(\vec{r}_i; \vec{\beta}_{-i})$$

$$p(\vec{r}_i; \vec{\beta}_{-i}) = \pi_{\text{int}}(\vec{r}_i; \vec{\beta}_{-i}) \cdot \vec{r}_i - \tau \log \left(\sum_{j=1}^{M} \exp\left(\frac{r_i(x, y_j)}{\tau} + \vec{\beta}_{-i,j}\right)\right) - C, \ C \in \mathbb{R} \tag{13}$$

$\square$

## B.2 DIFFERENCES FROM STANDARD AUCTION SETTINGS

Standard auction environments typically rely on a set of assumptions that simplify mechanism design; however, these assumptions do not apply to auctions for LLM-generated content. In this section, we detail these assumptions and discuss why they are inapplicable in our context.

First, in a standard auction setting, it is common to assume that the agents' valuation functions satisfy free disposal, i.e., $v_i(S) \geq v_i(S') \ \forall S \supseteq S', S, S' \supseteq \mathcal{I}$. The interpretation of free disposal is that an agent can discard any items she is allocated that she is not interested in. Free disposal combined with the fact that an agent has zero value for the empty bundle mean that her value for any outcome is weakly positive. Second, in most auction environments, the allocation rule is different for different agents: each agent will get allocated her own bundle of items, and we can assume that she is indifferent to the allocation of items to the other agents.

As detailed in Rafailov et al. (2023), assuming that an agent's LLM $\pi_i$ was trained to maximize her reward function (and regularized with respect to its KL divergence from some reference LLM, which we assume to be the same as the auctioneer's reference LLM), there is a one-to-many mapping between an advertiser's optimal LLM, and her implicit reward function. That mapping is:

$$r_i(x, y) = \tau_i \log \frac{\pi_i(y|x)}{\pi_{\text{ref}}(y|x)} + \log Z_i(x) \tag{14}$$

where $Z_i(x)$ is a prompt-dependent constant, and $\tau_i$ is the regularization hyperparameter of advertiser $i$, similar to the one in Equation (1). All functions in the class defined in Equation (14) are equivalent, in the sense that they induce exactly the same LLM (Rafailov et al., 2023). This has two implications: First, unlike standard auction environments, an agent's reward can go negative – there is nothing equivalent to the free disposal property. Setting $Z_i(x)$ to zero (which is equivalent to normalizing the induced probabilities by the LLM (Rafailov et al., 2023)), the agent's reward is negative for any sequence for which her LLM assigns a lower probability than $\pi_{\text{ref}}$.

Second, especially in the online advertising application, an agent's expected utility for not participating in the auction is negative: if advertiser $i$ does not participate in the auction, her payment is zero, but her expected value for the outcome is

$$\pi_{\theta_{-i,M}}(\vec{\beta}_{-i}) \cdot \vec{r_i} \tag{15}$$

The other advertisers have very low rewards for the sequences that mention advertiser $i$: assuming their LLMs have been properly trained, they will evaluate all sequences that explicitly mention a different, possibly competing brand, as unlikely. Thus, based on Equation (14) the corresponding advertisers have very low rewards for those sequences and conversely, advertiser $i$ has low rewards for the sequences that the other advertisers have high rewards for. But based on Equation (2), if advertiser $i$ does not participate in the auction, $\pi_{\theta_{-i,M}}(\vec{\beta}_{-i})$ will assign high probabilities to sequences for which $i$ has low rewards for. Thus, Equation (15) implies that, unlike standard auction environments, the advertiser's expected reward and utility for not participating in the mechanism is negative.

### B.3 OUR MECHANISM IS "ALMOST INDIVIDUALLY RATIONAL"

First, we explain why the standard notion of individual rationality (i.e., weakly positive utility from participation in the mechanism) encountered in most auction settings is impossible to achieve in this domain while converging to the optimal distribution and maintaining incentive compatibility. Then, we explain how, with our payment offset, our mechanism is "almost IR:" In Lemma B.1 we prove that the ex-interim utility of an advertiser who has zero reward for all candidate sequences and bids truthfully is deterministically zero, i.e., advertisers that do not contribute to the social welfare (but also do not detract from it) have zero utility. Similarly, in Lemma B.2 we prove that if an agent's reward for all candidate sequences is (weakly) positive, then her ex-interim utility is (weakly) positive.

**Why is indivual rationality (IR) impossible?** *Individual rationality* (IR) stipulates that an agent gains more utility by participating and bidding truthfully in a mechanism than by not participating at all. Typically, if an agent's utility for non-participation is zero, participating should yield weakly positive utility. However, this simplification does not apply in our setting.

As discussed in Section 5.2, advertiser $i$'s reward for any sequence $y$ can be arbitrarily negative (Equation (14)). The same is true for the utility from truthful participation, as outlined in Equation (12). To ensure a positive utility for every advertiser in our mechanism, an offset would need to be infinitely large or dependent on advertiser $i$'s reports. But then the mechanism's allocation rule would no longer be the gradient of advertiser $i$'s utility with respect to her reports, which would destroy strategyproofness (Frongillo & Kash, 2021; Rochet, 1987; Myerson, 1981).

It is important to note that this challenge is inherent not just to our mechanism but to any mechanism in this setting that operates with a fixed set of sequences, aims to approximate the optimal distribution, and maintains strategyproofness. Under these conditions, the only allocation rule that approximates the theoretically optimal distribution (Equation (2)) is that of our mechanism. However, this uniquely determines the advertisers' utilities, up to a constant factor, as described in Equation (12) (Frongillo & Kash, 2021; Rochet, 1987; Myerson, 1981).

**Lemma B.1.** *For the payment offset $C = -\tau \log \left( \sum_{j=1}^{M} \exp \left( \vec{\beta}_{-i,j} \right) \right)$ if advertiser $i$'s reward for all candidate sequences is zero, then her ex-interim utility is deterministically zero, for all $\vec{\beta}_{-i} \in \vec{R}_{-i}$.*

*Lemma B.1 Proof.* First, note that for all $\vec{\beta}_{-i} \in \vec{R}_{-i}$, advertiser $i$'s expected reward for the outcome is zero, as $\pi_{\text{int}}(\vec{r}_i; \vec{\beta}_{-i}) \cdot \vec{r}_i = \pi_{\text{int}}(\vec{r}_i; \vec{\beta}_{-i}) \cdot \vec{0} = 0$. Additionally, advertiser $i$'s reward for the realized outcome will deterministically be zero, as her reward for all generated candidate sequences is zero. Finally, note that by setting $\vec{r}_i = \vec{0}$ in Equation (13) with the offset $C$ set as in Section 5.3, we have that the advertiser $i$'s payment is also deterministically zero. Thus, an advertiser with zero reward for all generated candidate sequences who reports her rewards truthfully has deterministically zero reward for the final outcome and zero payments, and her utility is also deterministically zero.

□

**Lemma B.2.** *For the payment offset $C = -\tau \log \left( \sum_{j=1}^{M} \exp \left( \vec{\beta}_{-i,j} \right) \right)$ if advertiser $i$'s reward for all candidate sequences is positive, then her ex-interim utility is positive, for all reports $\vec{\beta}_{-i} \in \vec{R}_{-i}$.*

*Proof.* Lemma B.1 establishes that when advertiser $i$'s reward for all candidate sequences is zero, her utility for truthfully bidding in the mechanism, denoted as $U(\vec{0}; \vec{\beta}_{-i})$, is zero for all possible reports of the other advertisers $\vec{\beta}_{-i} \in \vec{R}_{-i}$.

Furthermore, Theorem 5.1 shows that the mechanism's allocation rule corresponds to the gradient of advertiser $i$'s utility when bidding truthfully. Because the allocation rule is non-negative, the gradient of advertiser $i$'s utility for bidding truthfully is also non-negative.

Thus, if advertiser $i$'s rewards for all candidate sequences are weakly positive, and considering the non-negative gradient of her utility, her ex-interim utility under truthful bidding must be positive, irrespective of the other advertisers' reports $\vec{\beta}_{-i}$.

□

**Corollary B.3.** *For the payment offset* $C = -\tau \log \left( \sum_{j=1}^{M} \exp \left( \vec{\beta}_{-i,j} \right) \right)$ *if the distribution* $\pi_{con}$ *only generates candidate sequences for which advertiser $i$'s reward is positive, then the ex-ante expected utility of the advertiser is positive.*

*Corollary B.3 Proof.* This follows immediately from the fact the the fact that the ex-ante utility of the advertiser is the expectation of her ex-interim utility with respect to her reward for the generated sequences, and the fact that the second quantity is positive whenever the reward of the advertiser for all candidate sequences is positive from Lemma B.2.

$\square$

### B.4 "What you give is what you get"

As we explained in Section 5.3.2, our allocation rule, which is the only one over a finite set of replies that converges to the optimal LLM, is also the (sub)gradient of the utility to ensure truthfulness (Rochet, 1987). Because the allocation rule is the same for all advertisers, their utilities must also be the same, up to advertiser-specific offsets, as indicated by Equation (16):

$$U_C(\vec{r}_i; \vec{\beta}_{-i}) = \tau \log \left( \sum_{j=1}^{M} \exp \left( \frac{1}{\tau} \sum_{k \in N} r_k(x, y_j) \right) + \log \frac{\pi_{\text{ref}}(y_j | x)}{\pi_{\text{con}}(y_j | x; \vec{c})} \right) + C, \; C \in \mathbb{R} \quad (16)$$

However, not all advertisers contribute equally to the social welfare of the final outcome. Because of this, implementing the mechanism without a carefully-designed offset would lead to free-riding: as long as an agent's utility in Equation (16) is positive, she would be incentivized to participate, even if the user query was completely unrelated to her business, because the mechanism would ensure that she received, on expectation, the same (positive) expected utility from doing so as any other participating advertiser.[6]

Incentivizing unrelated advertisers to participate would would have adverse effects. First, the better-performing context-aware mechanism would create candidate sequences with worse rewards for *all* advertisers, because its context would be "diluted" from advertisers unrelated to the user query. In our running example for the query "How to learn music online?", imagine adding "Try to mention 'EasySwitch', a comprehensive VPN service" to the context of the context-aware LLM $\pi_{\text{con}}(\cdot | x; \vec{c})$.

Additionally, for both versions of the mechanism, following the discussion in Section 5.2, the advertisers for whom the user query is unrelated are more likely to have negative rewards for the generated sequences as their LLMs will deem the candidate sequences more unlikely than the reference LLM. Thus, based on Equation (4), their participation in the mechanism will lead to a reduction of the total sum of rewards of the generated sequences for the advertisers, which will indirectly reduce the expected utility of all advertisers, making the mechanism less attractive for user-query-relevant advertisers.

To summarize, all advertisers receiving the same utility would incentivize advertisers for whom the user query is unrelated to participate in the auction. This would in turn reduce everyone's expected utility, potentially reducing the incentive for the user-query-relevant advertisers to participate, and lead to sequences with worse expected rewards for the advertisers and usefulness for the user.[7] Thus, in the application of auctions for aggregating advertisers' preferences over LLM-generated outputs, advertisers with higher contribution to social welfare also receiving proportionally higher utility by the mechanism is important for the long-term success of the mechanism in practice.

---

[6]We can assume that advertisers can estimate their expected utility from participation using historical data from past auctions, analogously to how they can estimate their utility for participating in sponsored search auctions.

[7]If we interpret the KL divergence between the distribution induced by the reference LLM and the LLM that generated the candidate sequences as a measure of their expected usefulness for the user.

## C   DETAILS FROM SECTION 6

### C.1   DETAILED EXPERIMENT SETUP

We create a set of synthetic instances to test our mechanism. Each instance consists of a user query, e.g. "How do I bake cookies?" and a list of advertisers. Each advertiser is defined by an "advertiser name", e.g. "KitchenFix" and an advertiser description, e.g., "producing kitchen appliances."[8] The reference LLM $\pi_{\text{ref}}$ responsible for generating replies that are useful for the user is Llama-2-7b-chat-hf (Touvron et al., 2023). The advertisers' LLMs are created using the same reference LLM, and adding the instruction: "Answer the question advertising ⟨advertiser⟩, ⟨advertiser description⟩." The context aware LLM is created using the same reference LLM, and adding the instruction: "Answer the query. Try to mention ⟨advertiser 1⟩, who ⟨advertiser description 1⟩  and ⟨advertiser 2⟩, who ⟨advertiser description 2⟩."

Following (Rafailov et al., 2023) the reward function of advertiser $i$ is set to $r_i(x, y) = \log \frac{\pi_i(y|x)}{\pi_{\text{ref}}(y|x)}$, where $\pi_i$ is advertiser $i$'s LLM, i.e., we set $\tau_i = 1, Z_i(x) = 1$ for all advertisers and for all user prompts in Equation (14).[9] For the auctioneer's objective as defined in Equation (1) we set $\tau = 1$, balancing between the advertisers' expected rewards for the generated sequences and the sequences' divergence from the reference LLM responsible for generating useful replies for the user. Thus, the optimal policy according to Equation (2) becomes:

$$\pi_r(y|x) = \frac{1}{Z(x)}\pi_{\text{ref}}(y|x)\exp\left(\sum_{i \in N} r_i(x, y)\right) \tag{17}$$

Following Li et al. (2024); Rozière et al. (2024) we sample from the LLM generating the sequences (either $\pi_{\text{ref}}$ or $\pi_{\text{con}}$) with temperature $0.8$ and top-p $0.95$. We use $50$ user queries, each with two interested advertisers. To increase the statistical significance of our results, we test each user query on $25$ different random seeds, effectively resulting in $1250$ individual instances. The seed affects both the generation of the candidate sequences, and the randomized sampling of the final sequence from the candidate sequences. We will make all of our code and set of synthetic instances publicly available.

All experiments were conducted on a compute cluster running Ubuntu 20.04.6 LTS with AMD EPYC processors with 48 cores and 1512GB RAM and Nvidia A100 GPUs and Python 3.12.1.

---

[8]Following Duetting et al. (2023), we use fictitious brand names to avoid the model using any existing ad.

[9]Note that we cannot estimate the true value of $Z_i(x)$. However, it is a query-dependent constant. Given that our experiments focus on the reward difference of different sequences for the same query, this is a non-issue.

## C.2    COMPARISON OF BASELINE AND CONTEXT-AWARE MECHANISM OUTPUTS

To provide some intuition of how the context-aware mechanism improves the relevance of generated responses for the advertisers, we present a side-by-side comparison of outputs from both mechanisms for a specific query in Table 2. Note that only the context-aware mechanism is able to incorporate the advertisers into the reply to the user's query.

| **User query** |
| --- |
| What are effective ways to learn a musical instrument online? |
| **Advertisers** |
| - MusicMastery: offering online music lessons and virtual tutoring
- InstaTune: selling musical instruments and learning aids |
| **Context-Aware Mechanism Output** |
| Effective ways to learn a musical instrument online include:
1. Online music lessons: Platforms like **MusicMastery** offer one-on-one virtual tutoring, group classes, and pre-recorded lessons. These lessons can be tailored to your skill level and learning style.
2. Interactive learning tools: **InstaTune** offers a range of interactive learning tools, such as apps, software, and games, that make learning fun and engaging.
3. Practice with technology: Utilize digital tools like metronomes, tuners, and recording software to practice and track your progress.
4. Collaborate with other musicians: Join online communities, forums, and social media groups to connect with other musicians and learn from their experiences. . . . |
| **Baseline Mechanism Output** |
| - Practice consistently and regularly: Consistent practice is essential to learning any musical instrument, and it's even more important when learning online. Set aside a specific time each day or week to practice, and make sure to stick to it.
- Find a good teacher or mentor: While it's possible to learn some things on your own, working with a good teacher or mentor can help you improve faster and avoid developing bad habits. Look for someone who is experienced and knowledgeable in the type of music you want to learn.
- Use technology to your advantage: There are many online tools and resources available that can help you learn an instrument. These include video lessons, online tutorials, and apps that can help you practice and track your progress. . . . |

Table 2: Comparison of context-aware and baseline mechanism outputs using 10 candidate sequences. Advertisers highlighted in bold, outputs truncated to conserve space.

C.3 EXPERIMENTAL EVALUATION OF THE USEFULNESS TO THE USER

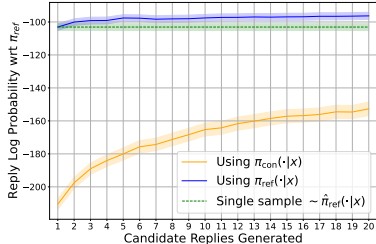

Figure 3: Reply log probability with respect to the reference LLM as a function of the number of replies generated using $\pi_{\text{ref}}$ and $\pi_{\text{con}}$.

In this section, we experimentally evaluate the usefulness of the replies returned by MOSAIC from the user's perspective.

In Figure 3, we plot the log probability of the replies returned by MOSAIC with respect to the optimal LLM for the user (i.e., the reference LLM) as a function of the number of candidate replies generated. We compare the context-aware version of MOSAIC to a baseline version. To benchmark our results, we also compare against sampling a *single reply* directly from the optimal distribution for this task, which is the reference LLM itself.

The results in Figure 3 show that for the context-aware version of our mechanism, the log probability of the replies relative to the reference LLM increases rapidly as the number of candidate replies grows. This demonstrates that by generating more replies, the context-aware version of MOSAIC is able to produce replies with substantial value for the user. By comparison, the usefulness to the user that the baseline version is able to achieve does not scale with more generated replies.

However, there remains a gap between the log probability of those replies and the benchmark set by the reference LLM. This discrepancy arises because the platform optimizes a different objective: the expected advertiser rewards combined with the KL divergence from the reference LLM. By increasing the weight $\tau$ that the platform places on the reference LLM, this gap between the context-aware version of MOSAIC and the user's optimal benchmark will close.

C.4 COMPREHENSIVE EXPERIMENTAL EVALUATION OF THE OFFSET FROM SECTION 5.3

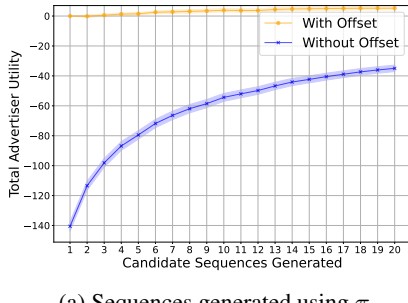 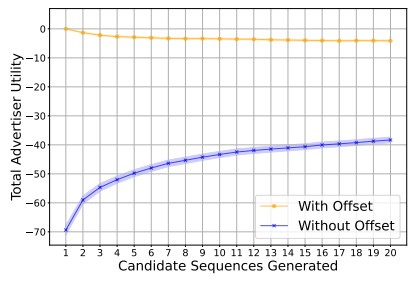

(a) Sequences generated using $\pi_{\text{con}}$        (b) Sequences generated using $\pi_{\text{ref}}$

Figure 4: Comparison of total advertiser utility gain from participation with, and without the payment offset, as a function of the number of candidate sequences generated using $\pi_{\text{ref}}$ and $\pi_{\text{con}}$. Averaged over 1250 runs including 95% CIs.

In this section, we provide a comprehensive experimental evaluation of the payment offset introduced in Section 5.3. Figure 4 explores the effectiveness of the payment offset introduced in Section 5.3 in ensuring that our mechanism is ex-ante IR, i.e., advertisers have positive expected utility gains from participating. To that end, we compare the advertiser utility gain from participation with and

without the payment offset, as a function of the number of candidate sequences generated by the context-aware (Figure 4a) and baseline (Figure 4b) versions of our mechanism.

In Figure 4a we observe that for the context-aware version of our mechanism, adding the payment offset introduced of Section 5.3 to the payment rule is enough to ensure positive expected utility for the advertisers (conditioned on the fact that they are related to the user's query), i.e., make the mechanism ex-ante IR. Notably, without the payment offset, advertisers have very negative expected utility from participating in the mechanism.

In Figure 4b we observe that for the baseline version of our mechanism, adding the payment offset introduced of Section 5.3 to the payment rule causes a very large increase in the advertisers' expected utility, but it is still not enough to ensure positive expected utility for the advertisers. To conclude, Figure 4 demonstrates that the offset introduced in Section 5.3 causes a large increase in the advertisers' expected utility, which in case of the context-aware version of the mechanism, is also enough to make the mechanism ex-ante IR.

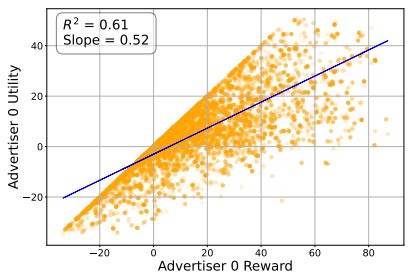

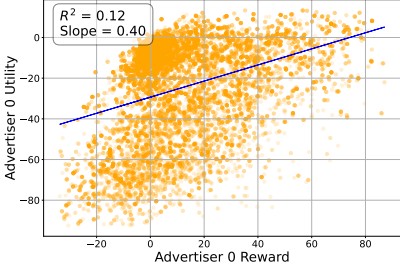

(a) With the payment offset.

(b) Without the payment offset.

Figure 5: Comparative scatter plots of advertiser reward and utility gain from participation, with and without the payment offset of Section 5.3 for candidate sequences generated by the context-aware LLM $\pi_{\text{gem}}$. We additionally show a linear regressor fit to that data, its slope and its $R^2$.

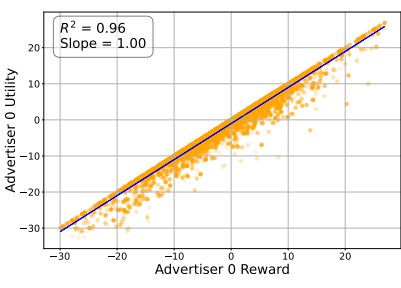

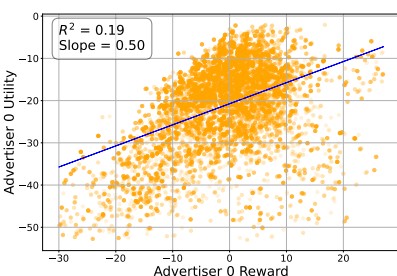

(a) With the payment offset.

(b) Without the payment offset.

Figure 6: Comparative scatter plots of advertiser reward and utility gain from participation, with and without the payment offset of Section 5.3 for candidate sequences generated by the reference $\pi_{\text{ref}}$. We additionally show a linear regressor fit to that data, its slope and its $R^2$.

Figures 5 and 6 explore the effectiveness of the payment offset introduced in Section 5.3 in aligning an advertiser's utility with her contribution to the social welfare. In Figure 5 we compare the scatter plots of the advertiser reward and utility gain from participation in the mechanism, with and without the payment offset introduced in Section 5.3 for candidate sequences generated using the context-aware LLM $\pi_{\text{ref}}$. Additionally, for both subfigures, we show a linear regressor fitted to the data, as well as its slope and coefficient of determination. Comparing the two subfigures, it is immediately obvious that adding the offset to the payments makes the relationship between advertiser reward and utility gain far more linear. This is confirmed by the coefficient of determination of the linear regressors fit to each dataset. The coefficient of determination of the linear regressor is far larger when we use the offset. Without the payment offset, the coefficient of determination is almost 0, indicating that,

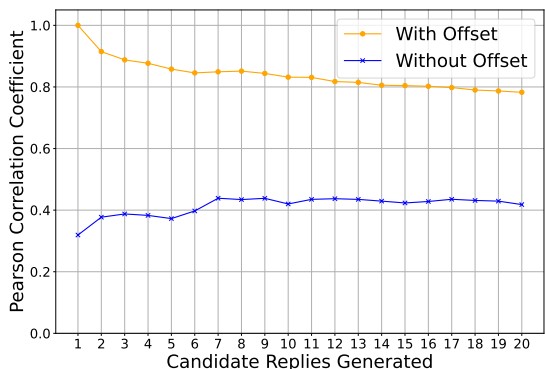

Figure 7: Pearson correlation between advertiser normalized utility and reward.

without our payment offset, reward gain is not a predictive measure of an agent's utility. Additionally, the slope of the linear regressor is also higher for the scatter plot with the payment offset.

In Figure 6 we make the same comparison, but for candidate sequences generated using the reference LLM $\pi_{\text{ref}}$. The results are now even more pronounced. In Figure 6a we observe the relationship between advertiser utility and reward gain with our payment offset is almost perfectly linear, as suggested by the linear regressor fitted to the data having a slope of $1.00$ and an extremely high coefficient of determination of $0.96$, indicating that it can almost perfectly fit the data. Without our payment offset however, in Figure 6b we can see that the relationship between the two metrics is again both less linear, and less positively correlated, as the slope of the linear regressor is $0.5$ and its coefficient of determination is only $0.19$.

To conclude, in all cases tested, the use of the advertiser-specific offset introduced in Section 5.3 increases an advertiser's expected utility, makes the relationship between an advertiser's contribution to social welfare and her utility both more linear and more positively correlated, and, for the context-aware version of our mechanism, can ensure positive expected utility for the advertisers.

# D  ADDITIONAL INFORMATION FOR ICLR REVIEWERS

In this section we include additional information addressing certain points raised by the conference reviewers.

## D.1  ADDITIONAL TECHNICAL CONSIDERATIONS

**Output Constraints.**   The generation of candidate replies is performed by $\pi_{\text{gen}}$, which is controlled by the platform. Thus, MOSAIC allows the platform to incorporate any desired constraints such as maximum length on the output replies by adding those constraints to the candidate reply generation procedure.

## D.2  BUDGET AND ROI CONSTRAINTS

MOSAIC supports both budget and ROI constraints. For budget constraints, the advertiser should only bid rewards up to her maximum budget. For ROI constraints, the advertiser should "bid shade" when reporting her reward function to ensure that her ROI constraint is satisfied. As in any other strategyproof mechanism, both of these misreports would lead to worse utility per auction instance for the advertiser compared to truthful bidding, because in MOSAIC it is a dominant strategy for an advertiser to report truthfully. However, the upshot for the advertiser is that she can extend her budget to more auction instances. Additionally, for MOSAIC we can prove a new version of the "what you give is what you get" property of Lemma 5.2 for the case when a bidder under-reports her rewards. Informally, if an advertiser's under-reported rewards are still positive for all replies, then her utility when under-reporting will be positive, and higher than Lemma 5.2 would suggest. The formal lemma follows below:

**Lemma D.1.** *Let $\vec{r}_i$ be the true rewards of advertiser $i$ for the candidate replies, and $\vec{r}_i'$ be her reported rewards, where $\vec{r}_{i,j} \geq \vec{r}_{i,j}'$ for all $j \in [M]$. Finally, let $U_i(\vec{r}_i'; \vec{r}_i, \vec{\beta}_{-i})$ be advertiser $i$'s utility when her true rewards are $\vec{r}_i$ and she reports $\vec{r}_i'$. Then, for advertiser $i$'s utility under reports $\vec{r}_i$ and $\vec{r'}$ we have:*

$$U_i(\vec{r}_i; \vec{r}_i, \vec{\beta}_{-i}) \geq U_i(\vec{r}_i'; \vec{r}_i, \vec{\beta}_{-i}) \geq U_i(\vec{r}_i'; \vec{r}_i', \vec{\beta}_{-i}). \tag{18}$$

*In particular, this means that if $\vec{r} \geq \vec{r}_i' \geq \vec{0}$, then her utility will still be positive if she under-reports $\vec{r}_i'$ instead of $\vec{r}_i$.*

*Proof.*  The first inequality follows immediately from the fact that MOSAIC is truthful, i.e., it is a dominant strategy for advertiser $i$ to truthfully report her rewards $\vec{r}_i$. Thus, any misreport will lead to lower utility.

For the second inequality, note that both advertiser $i$'s payments and the allocation rule only depend on the advertiser's reports, not on her true preferences. Thus, both advertiser $i$'s payments and the probability of returning each of the candidate replies under the interim allocation rule are the same when advertiser $i$ reports $\vec{r}_i'$, regardless of whether her true rewards are $\vec{r}_i'$ or $\vec{r}_i$.

But for advertiser $i$'s reward in the two cases, we have:

$$\sum_{j=1}^{M} r_i(x, y_j)\pi_{\text{int}}(y_j|x; \vec{r}_{-i}, \vec{r}_i')_j \geq \sum_{j=1}^{M} r_i'(x, y_j)\pi_{\text{int}}(y_j|x; \vec{r}_{-i}, \vec{r}_i')_j \tag{19}$$

where the inequality follows because $r_i(x, y_j) \geq r_i'(x, y_j)$ for all $j \in [M]$.

Thus, when the advertiser has reward $\vec{r}_i$ but reports $\vec{r}_i' \leq \vec{r}_i$, she has the same payment as if her true rewards were $\vec{r}_i'$, but she has higher expected reward for the outcome, and thus higher utility, i.e. $U_i(\vec{r}_i'; \vec{r}_i, \vec{\beta}_{-i}) \geq U_i(\vec{r}_i'; \vec{r}_i', \vec{\beta}_{-i})$.

Finally, note that if an advertiser's true rewards for all replies is zero, her utility is zero from Lemma 5.2. Thus, if $\vec{r}_i' \geq \vec{0}$, we have $U_i(\vec{r}_i'; \vec{r}_i', \vec{\beta}_{-i}) \geq 0$. This concludes the proof.

$\square$

### D.3 CONFLICTS BETWEEN ADVERTISERS

The output of MOSAIC is a single text response to the user. In trying to reflect the interests of multiple advertisers, there is a risk of a single response being incoherent, just mentioning as many advertisers as possible without being a good promotion for any of them. However, the platform's objective, which takes into account both advertiser utility and usefulness for the user, should account for this: if advertisers are happy to be cross-promoted, those outcomes should be more likely, but incoherent advertisements or those that simultaneously promote competitors should be less likely.

In Figure 8 we show the fractions of advertisers mentioned by the context-aware version of MOSAIC as a function of the candidate number of replies used. We observe that initially, the fraction of advertisers mentioned is very high, but drops significantly once the mechanism has converged. The reason is precisely the fact that mentioning more advertisers in this case would cause either a significant drop in advertiser utility, or a drop in the usefulness of the reply to the user. By placing more weight towards the advertisers (i.e., decreasing the value of $\tau$ in Equation (1)), the platform can increase the number of advertisers shown when the mechanism has converged.

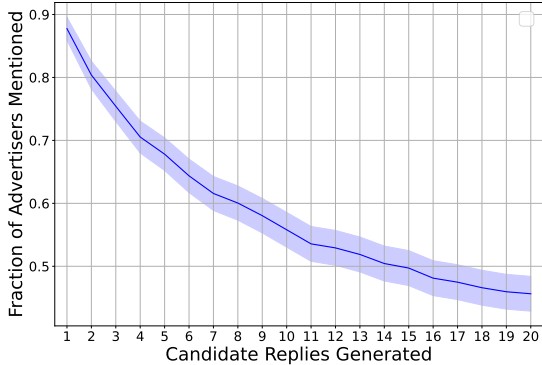

Figure 8: Fraction of advertisers mentioned in the output reply as a function of the number of candidate replies.

### D.4 TESTING WITH ADDITIONAL LANGUAGE MODELS

In this section, we test MOSAIC's performance using Google's flan-t5-large model (Chung et al., 2022).

We repeat exactly the same procedure as detailed in Section 6.1, but using flan-t5-large as the basis for both $\pi_{\text{gen}}$, the advertisers' LLMs, and the reference LLM. We use the same 50 user queries, each with two advertisers, and test each query on 10 different random seeds, resulting in 500 instances.

In Figure 9 to Figure 12, we present the total advertiser value gain, reward gain from participating in MOSAIC, platform revenue, and the usefulness of the returned reply to users as functions of the number of candidate replies generated.

Consistent with the main paper's results using Llama-2-7b-chat-hf (Touvron et al., 2023), the context-aware version of MOSAIC demonstrates significant value and utility generation for advertisers and successfully captures a substantial portion as platform revenue. In contrast, though, the naive version of MOSAIC fails to generate value or utility for advertisers, resulting in negligible revenue for the platform. Similar to the results of the main paper, despite both mechanisms converging in the limit to the platform's objective, as shown in Corollary 4.1, their performance for a computationally tractable number of generated replies diverges.

Similar to the findings in the main paper, the naive version retains the reference LLM's usefulness to the user (which is optimal for the task), while the context-aware version prioritizes the platform's objective, leading to lower user usefulness. Overall, our results using flan-t5-large agree closely with those obtained using Llama-2-7b-chat-hf, demonstrating consistency across models. The main difference is that the context-aware mechanism is even more important when using flan-t5-large.

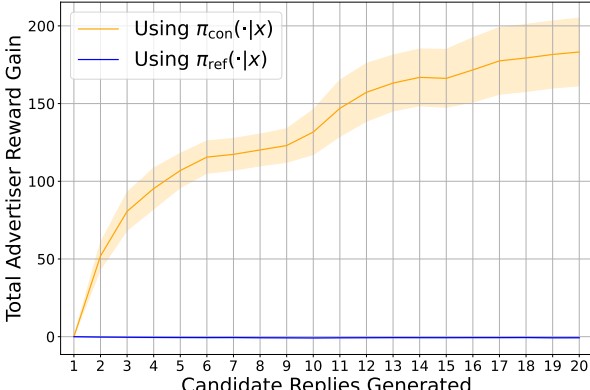

Figure 9: Advertiser value gain as a function of the number of candidate replies generated using $\pi_{\text{ref}}$ and $\pi_{\text{con}}$ (flan-t5-large model).

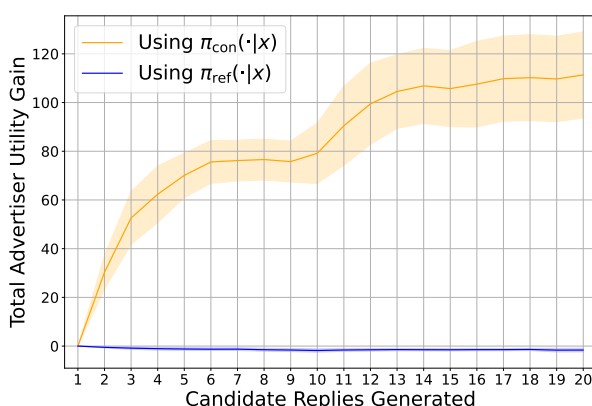

Figure 10: Advertiser value gain as a function of the number of candidate replies generated using $\pi_{\text{ref}}$ and $\pi_{\text{con}}$ (flan-t5-large model).

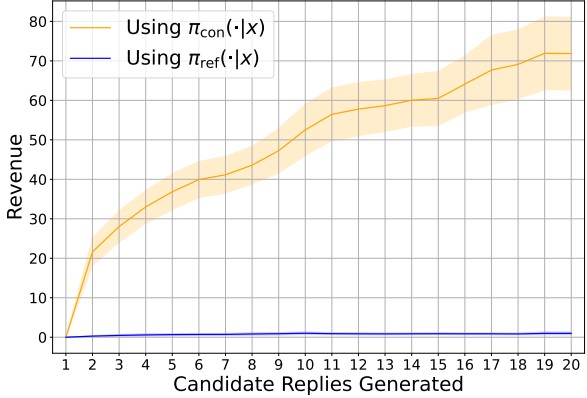

Figure 11: Revenue as a function of the number of candidate replies generated using $\pi_{\text{ref}}$ and $\pi_{\text{con}}$ (flan-t5-large model).

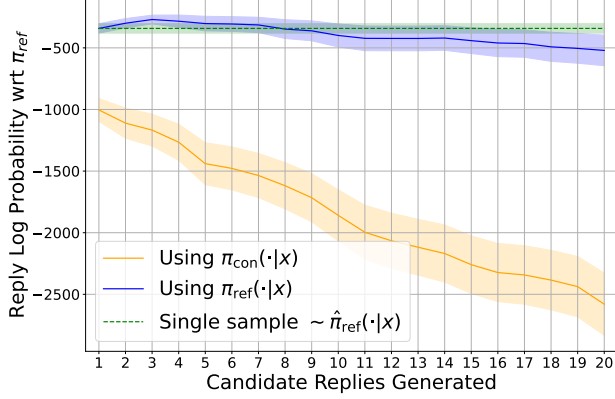

Figure 12: Reply log probability with respect to the reference LLM as a function of the number of candidate replies generated using $\pi_{\text{ref}}$ and $\pi_{\text{con}}$ (flan-t5-large model).

# E    FULL TEXT OF PROMPTS

Table 3: Table of user prompts with corresponding advertisers and their descriptions. Only the first two advertisers are shown for each query; the full prompts are available in the code supplement.

**User Query 1**
Learning to play the guitar.
**Advertisers**
- StrumMaster: selling guitars, tuners, and learning materials for beginners
- Chordify: offering online guitar lessons and interactive practice tools

**User Query 2**
Best skin care routine for dry skin.
**Advertisers**
- HydraSkin: providing moisturizers and serums for dry skin
- GlowEssentials: selling natural and organic skin care products

**User Query 3**
Improving productivity while working from home.
**Advertisers**
- SpaceMakers: offering ergonomic furniture and home office setups
- FocusApp: providing productivity apps and time management tools

**User Query 4**
Starting a small online business.
**Advertisers**
- EcomLaunch: offering e-commerce platform solutions and web design services
- MarketMover: providing digital marketing services and SEO optimization

**User Query 5**
Healthy meal planning on a budget.
**Advertisers**
- BudgetBites: selling affordable meal kits and recipe books
- NutriSaver: offering discounts on healthy groceries and food delivery services

**User Query 6**
Mastering digital photography.
**Advertisers**
- PixelPro: selling cameras, lenses, and photography accessories
- EditCraft: offering photo editing software and online tutorials

**User Query 7**
Effective ways to reduce household energy use.
**Advertisers**
- EcoSave: offering energy-efficient home appliances and lighting solutions
- InsulaTech: providing home insulation and energy audit services

**User Query 8**
Finding the perfect hiking trails.
**Advertisers**
- TrailFinder: offering a mobile app with detailed maps and trail reviews
- GearUp: selling outdoor gear and apparel for hiking enthusiasts

**User Query 9**
Building a personal brand on social media.
**Advertisers**
- BrandBuilder: offering personal branding courses and social media strategy consultations
- VisualizeMe: providing graphic design services for social media content

**User Query 10**
Learning a new language effectively.
**Advertisers**
- LingoLeap: offering online language learning courses and tutoring

- SpeakEasy: providing language learning apps with speech recognition technology

**User Query 11**
Staying fit without a gym.
**Advertisers**
- HomeFit: selling home workout equipment and fitness accessories
- MoveIt: offering online fitness classes and personal training sessions

**User Query 12**
Eco-friendly travel options.
**Advertisers**
- GreenPath: offering eco-friendly travel packages and sustainable tourism experiences
- EcoStay: providing listings for green hotels and accommodations

**User Query 13**
Mastering the art of cooking steak.
**Advertisers**
- GrillMaster: selling premium grills and barbecue accessories
- SteakPerfection: offering online cooking classes focused on meat preparation

**User Query 14**
Creating a successful YouTube channel.
**Advertisers**
- VidGrowth: offering video production courses and YouTube growth strategies
- ChannelDesign: providing custom YouTube channel art and video thumbnails

**User Query 15**
Decorating your home on a budget.
**Advertisers**
- DecorDeals: selling affordable home decor and furniture
- StyleSavvy: offering interior design consultations and budget-friendly decorating tips

**User Query 16**
Managing stress and anxiety.
**Advertisers**
- CalmSpace: offering mindfulness apps and stress reduction tools
- WellnessWave: providing online therapy sessions and mental health resources

**User Query 17**
How can I make my home more energy efficient?
**Advertisers**
- EcoWise: offering smart home energy solutions and solar panels
- GreenInsulate: providing eco-friendly insulation materials and installation services

**User Query 18**
What are the best strategies for investing in the stock market?
**Advertisers**
- InvestSmart: offering investment advice and financial planning services
- StockGenius: providing an AI-powered stock market analysis tool

**User Query 19**
How can I improve my mental health?
**Advertisers**
- MindBalance: offering mental health apps and mindfulness courses
- SerenityNow: providing online counseling and therapy sessions

**User Query 20**
Where can I find organic and sustainable clothing?
**Advertisers**
- EcoWear: selling sustainable and organic clothing options
- GreenThreads: offering eco-friendly fabrics and fair-trade fashion

**User Query 21**
How do I start a vegetable garden in my backyard?

**Advertisers**
- GardenGuru: selling gardening tools, seeds, and starter kits
- PlantPerfect: offering online gardening courses and personalized garden planning

**User Query 22**
What are the best tools for remote team collaboration?
**Advertisers**
- TeamSync: offering collaboration software and project management tools
- ConnectSuite: providing video conferencing solutions and team communication platforms

**User Query 23**
How can I learn to cook healthy meals quickly?
**Advertisers**
- QuickChef: offering quick and healthy cooking classes online
- NutriFast: selling kitchen gadgets and appliances for healthy cooking

**User Query 24**
What are the best ways to save for retirement?
**Advertisers**
- FutureFund: offering retirement planning and investment services
- PensionPlus: providing personalized financial advice for retirement savings

**User Query 25**
How can I decorate my apartment on a minimalist budget?
**Advertisers**
- MinimalSpace: selling minimalist furniture and decor
- SimplicityDesigns: offering interior design consultation for minimalist living spaces

**User Query 26**
What are the best practices for data security in small businesses?
**Advertisers**
- SecureIT: offering cybersecurity solutions for small businesses
- DataGuard: providing data protection services and training

**User Query 27**
How can I reduce my carbon footprint with daily habits?
**Advertisers**
- EcoHabits: offering products and tips for sustainable living
- GreenStep: providing carbon offset subscriptions and eco-friendly lifestyle goods

**User Query 28**
What are effective ways to learn a musical instrument online?
**Advertisers**
- MusicMastery: offering online music lessons and virtual tutoring
- InstaTune: selling musical instruments and learning aids

**User Query 29**
How can I keep my pets healthy and happy?
**Advertisers**
- PetWell: offering pet health insurance and wellness products
- HappyPaws: selling organic pet food and eco-friendly toys

**User Query 30**
What's the best way to start learning coding from scratch?
**Advertisers**
- CodeBegin: offering coding bootcamps and online tutorials for beginners
- DevPath: providing interactive coding courses and learning platforms

**User Query 31**
How to find affordable eco-friendly travel destinations?
**Advertisers**
- GreenTraveler: offering eco-friendly travel guides and affordable green vacation packages
- EcoTripAdvisor: providing reviews and bookings for sustainable travel options

**User Query 32**
What are the best ways to improve indoor air quality?
**Advertisers**
- AirPure: selling air purifiers and natural air freshening solutions
- BreatheEasy: offering indoor plant subscriptions and ventilation solutions

**User Query 33**
How to make my own natural beauty products?
**Advertisers**
- NatureGlow: offering DIY beauty product kits and organic ingredients
- EcoBeauty: selling guides and supplies for creating homemade natural cosmetics

**User Query 34**
What are the best techniques for effective online teaching?
**Advertisers**
- EduTech: offering online teaching tools and educational technology solutions
- TeachMaster: providing courses on online pedagogy and digital classroom management

**User Query 35**
How to create a successful podcast from home?
**Advertisers**
- PodCastify: offering podcast creation courses and audio equipment
- SoundWave: selling podcast editing software and hosting services

**User Query 36**
What are the most effective ways to market a small business locally?
**Advertisers**
- LocalBoost: offering local SEO services and marketing strategies for small businesses
- MarketTown: providing targeted advertising solutions and local business networking opportunities

**User Query 37**
How can I optimize my home office for productivity?
**Advertisers**
- WorkZen: offering ergonomic office furniture and productivity tools
- SpaceCraft: providing home office design services and organization solutions

**User Query 38**
What are the best platforms for freelance writers to find work?
**Advertisers**
- WriteFreely: offering a job board and resources for freelance writers
- ContentConnect: providing a platform to connect writers with businesses

**User Query 39**
How to get started with meditation and mindfulness?
**Advertisers**
- MindfulStart: offering guided meditation apps and mindfulness training
- ZenSpace: selling meditation cushions and mindfulness journals

**User Query 40**
Where can I find unique handmade gifts?
**Advertisers**
- CraftCorner: selling handmade gifts and artisan crafts
- MakerMarket: offering a marketplace for unique handmade items from local artists

**User Query 41**
How can I improve my credit score quickly?
**Advertisers**
- CreditBoost: offering credit monitoring services and financial advice
- ScoreSurge: providing tools and tips for improving your credit score

**User Query 42**
What are the best apps for tracking fitness and nutrition?
**Advertisers**
- FitTrack: offering comprehensive fitness and nutrition tracking apps

- NutriSync: selling personalized nutrition plans and diet tracking tools

**User Query 43**
How to plan an environmentally friendly wedding?
**Advertisers**
- GreenBride: offering eco-friendly wedding planning services and supplies
- EcoWed: providing sustainable wedding attire and decor

**User Query 44**
What are the safest ways to travel during a pandemic?
**Advertisers**
- SafeJourney: offering travel safety kits and pandemic travel advice
- HealthTravel: providing information on safe destinations and travel insurance

**User Query 45**
How can I learn DIY home repairs and improvements?
**Advertisers**
- FixItYourself: offering online courses and tutorials for home repair
- DIYHome: selling DIY home improvement tools and kits

**User Query 46**
What are the best resources for starting a plant-based diet?
**Advertisers**
- PlantEats: offering guides and meal plans for starting a plant-based diet
- VeggieVibe: selling plant-based cookbooks and kitchen gadgets

**User Query 47**
Tips for running a marathon.
**Advertisers**
- RunFastGear: specializing in high-performance running shoes
- HydraFuel: offering electrolyte drinks and energy bars for athletes

**User Query 48**
Best practices for organic gardening.
**Advertisers**
- GreenThumb Solutions: providing organic fertilizers and pest control products
- EcoGrow: selling heirloom seeds and sustainable gardening tools

**User Query 49**
How to improve home WiFi?
**Advertisers**
- SignalBoost: which offers advanced WiFi routers and extenders
- NetWizard: providing network optimization services and support

**User Query 50**
Ways to save on travel.
**Advertisers**
- BudgetJourneys: specializing in affordable travel packages and deals
- StayLocal: offering discounts on boutique hotels and unique accommodations

