# OpenReview forum: "Truthful Aggregation of LLMs with an Application to Online Advertising"
_ICLR.cc/2025/Conference — Submitted to ICLR 2025_

### Official Review · Reviewer_MGd7 · 2024-10-30

**Soundness:** 2
**Presentation:** 3
**Contribution:** 2
**Rating:** 5
**Confidence:** 3

**Summary:**

This paper introduces MOSAIC, an auction mechanism designed for aggregating advertiser preferences within the outputs of large language models (LLMs), particularly for applications in online advertising. The authors address the challenge of balancing the interests of multiple advertisers with user-centric content by creating a mechanism that incentivizes truthful reporting from advertisers. MOSAIC uses an approach that combines an allocation rule based on importance sampling, allowing it to converge to the optimal output distribution as computational resources increase, without requiring direct model fine-tuning.

**Strengths:**

The paper introduces MOSAIC which effectively combines the preferences of self-interested advertisers while maintaining user-centric content. This is achieved without requiring direct access to LLM weights, which adds flexibility and broad applicability.

The mechanism ensures that truthful reporting is a dominant strategy for advertisers, thanks to its carefully constructed payment and allocation rules. This strategy-proofness is backed by theoretical proofs.

The design is computationally efficient, using only API access and avoiding expensive fine-tuning.

**Weaknesses:**

1. There is no real-world testing or application with actual advertiser data to validate the effectiveness of MOSAIC in practical applications.

2. Although MOSAIC is designed to be efficient, it may still face scalability challenges as the number of advertisers or candidate replies grows. An analysis of how the mechanism handles these scenarios under different computational limits would strengthen the paper.

3. Given that the mechanism optimizes for advertiser rewards and alignment with a reference LLM, there’s a risk of unintentional bias in the final output.

4. MOSAIC relies on advertisers truthfully reporting their preferences and reward functions, yet it does not fully address how misreporting could impact results. Further discussion on mid-reporting would be important.

**Questions:**

1. The paper claims that MOSAIC’s allocation rule converges to an optimal distribution as computational resources increase. How does the rate of convergence depend on the number of candidate sequences?

2. The allocation rule is based on importance sampling to estimate the probability distribution over sequences. What is the mathematical form of the variance for this estimator?

3. How does the choice of the hyperparameter \tau in front of the KL term quantitatively influence the trade-off between advertiser reward maximization and alignment with the reference LLM?

4. The strategyproofness guarantee hinges on honest reporting by advertisers, but how robust is MOSAIC if advertisers engage in complex forms of gaming or misreporting?

5.  In scenarios with advertisers holding strongly opposing interests, does MOSAIC risk generating responses that conflict with each other?

---

> ### Author Response · Authors · 2024-11-14
>
> Thank you for the review. Below are our responses:
>
>
> **"No real-world testing with actual advertiser data."** This is a great suggestion and we would really like to do it. The problem is that any such data would be closely held by advertising companies, and we don’t have access. As far as we are aware, there is no such public data available.
>
>
> **“An analysis of how the mechanism handles scalability under different numbers of candidate replies or advertisers:”** In Section 4.2, we explain why we consider our mechanism computationally efficient. Our mechanism scales linearly in both the number of participating advertisers and the number of candidate replies. This is also the case for the first and second price auctions that are used by major online platforms to run hundreds of billions of ad auctions on a yearly basis. We will expand on this section in the final version of the paper.
>
>
> **"Given that the mechanism optimizes for advertiser reward and divergence from a reference LLM, there is a risk of unintentional bias in the final output.”** In Section 4.1 and Appendix A.2, we prove that MOSAIC is an unbiased estimator of the optimal distribution for the platform’s objective. If the platform’s objective is unbiased, the same will be true for the output of our estimator. If bias here refers to societal bias, then using a base LLM which has been post-trained to remove such bias should help mitigate this issue.
>
> **"What is the mathematical variance of the estimator?**" The precise variance is provided in Section 4.1, lines 239-245, with a detailed derivation in Appendix A.2.
>
> **"Convergence rate with number of candidate sequences."** In section 6.2 figure (a), we show experimentally that MOSAIC can converge to the target distribution using 20 candidate replies. In Section 4. and appendix A.2, we show MOSAIC’s variance as an estimator. From the form of that variance, using standard concentration inequalities we can prove that our estimator converges to the true distribution at a rate proportional to 1$/\sqrt{M}$, where $M$ is the number of candidate replies generrated. We will add this detailed derivation in the appendix.
>
>
> **"How does the choice of hyperparameter $\tau$ qualitatively influence the trade-off between advertisers’ rewards and divergence from the reference LLM?"** The reference LLM can be interpreted as one additional advertiser with her own reward function. By setting $\tau$ equal to 1, all participants (the reference LLM and the advertisers) are treated equally. By setting $\tau$ to any other number, it is as though we insert $\tau$ such identical “reference LLM participants” in the auction. Thus, larger values of tau place more weight towards the reference LLM, and smaller values place more value towards the advertisers.
>
> **"The strategy proofness guarantee hinges on honest reporting by advertisers. But how robust is MOSAIC if advertisers engage in complex forms of gaming?"** MOSAIC is **dominant** strategy incentive compatible. This means that it is always in an advertiser’s best interests to report truthfully her preferences, even if the other advertisers are misreporting. There are no complex forms of gaming that will result in better outcomes for the advertisers – at least not unilaterally. This is one of MOSAIC’s greatest strengths. If advertisers are able to collude with each other, they might be able to benefit themselves, but this is an unavoidable problem that always arises in strategyproof mechanism design, and platforms can prevent this possibility in practice.
>
>
> **"In scenarios where advertisers have strongly opposing interests, does MOSAIC risk generating responses that conflict with each other?"** Potentially yes, but the platform’s objective can account for this. Intuitively, if the total utility of an "equivocating" answer for advertiser A and advertiser B is less than either "just prefer A", or "just prefer B", a single advertiser is more likely to be picked. On the other hand, if the advertisers are happy to be promoted together, that outcome is more likely. In our experiments, MOSAIC only includes 54% of the advertisers on average when it has converged, as opposed to 81% prior to convergence. We will include this result in the appendix.

---

### Official Review · Reviewer_gyNr · 2024-10-31

**Soundness:** 3
**Presentation:** 3
**Contribution:** 3
**Rating:** 6
**Confidence:** 2

**Summary:**

In this paper, the authors studied a setting in which the advertisers would like to influence the LLMs to generate their preferred contents and the platform would need to satisfy both advertiser preference and user utilities. The authors proposed an auction mechanism called MOSAIC that enjoys a number of advantages, including ensuring that truthful reporting is a dominant strategy for advertisers. The authors also suggested that the proposed mechanism is equipped with technical feasibility and practicality. To validate their claims, the authors provided both theoretical results and numerical experiments with a publicly available LLM.

**Strengths:**

- The paper is well written and organized.
- LLM and its application to online advertising, especially the domain of ad auctions, is a novel area to be studied with practical relevance.
- The theoretical results provided appear sound. The authors also clearly stated the difference between their proposed mechanism and previous auction mechanisms such as VCG.
- The numerical experiments (and the motivating example) provide interesting insights.

**Weaknesses:**

- I wonder if the authors can provide more explanations for why the ref LLM is not performing as well as the context-aware LLM. I understand that the authors have provided some intuitive explanations in Sections 4, but the ref LLM still appears to be an intuitive choice based on Corollary 4.1. Would it be possible to gain more insights into this difference from a theoretical point of view? Does the authors have any preliminary insights for how to introduce contextual information into the current model?
- The current framework does not consider a number of constraints that could impact the advertisers' decision-making process, such as each advertiser's budget and/or ROI constraints, or the maximum length of the LLM output. I wonder if the authors can comment on whether their mechanism can incorporate any of the above features.

**Questions:**

See weaknesses.

---

> ### Author Response · Authors · 2024-11-14
>
> Thank you for the helpful review. Below are our responses:
>
>
> **"Differences between reference and context-aware LLM from a theoretical perspective."** In Section 4.1, lines 239-246, we explain how our allocation rule is an importance-sampling based estimator, and show how the variance of that estimator depends on how well the LLM that generates the candidate replies “matches” with the reference LLM. If that LLM is not tuned well, the variance of the estimator will blow up, meaning it will require more samples to achieve the same quality of approximation. We provide all details in Appendix A.2.
>
>
> **"Do the authors have any preliminary insights for how to introduce contextual information into the current model?"** This depends on the application. For our flagship application of online advertising, we detail how we do this in the footnote of line 269. For 50 practical examples, please see the file advertiser_prompts.py provided in the supplementary material. We will also include these examples in the appendix for convenience.
>
>
> **“Can the mechanism incorporate constraints such as advertisers’ budget/ROI constraints, or the maximum length of the LLM output?”:** Yes, MOSAIC can incorporate all three of these considerations. The maximum length of the output reply can be easily incorporated by the platform. The platform chooses the LLM that generates the candidate replies, and it can enforce that constraint in the reply generation. For budget constraints, the advertiser should only bid rewards up to her maximum budget. For ROI constraints, the advertiser should “bid shade” when reporting her reward function to ensure that their ROI constraint is satisfied. Both of these changes would lead to worse utility per instance for the advertiser compared to truthful bidding. This is the case for any strategyproof mechanism, with the upshot being that she can extend her budget to more auction instances. However, for our mechanism we can additionally prove a new version of the “what you give is what you get” property (equation (5)) for this case. Informally: if an advertiser’s shaded reported rewards still contribute to the social welfare of the final allocation, then her true utility will be positive, and even higher than Lemma 5.2 in the current version would suggest. We will add this discussion and the new theoretical result in the appendix.

---

### Official Review · Reviewer_N7ft · 2024-11-03

**Soundness:** 2
**Presentation:** 2
**Contribution:** 1
**Rating:** 5
**Confidence:** 4

**Summary:**

This paper proposes a mechanism called MOSAIC, that aggregates multiple advertising LLMs (represented by reward functions) and try to find a distribution on replies that maximizes the total rewards of advertisers together with a KL-divergence regularization term on user preference $\pi_{ref}$. MOSAIC takes the reward functions as input, and output a (stochastic) reply to the user.

MOSAIC first samples $M$ candidate replies from arbitrary general and pre-defined distribution $\pi_{gen}$. After this, MOSAIC computes the distribution on replies (regarding the distributions as mechanism allocation) and sample one reply from this distribution as final output. The mechanism payments are computed by Rochet payment (1987). As $M$ tends to infinity, MOSAIC is guaranteed to converge to VCG mechanism.

In experiments, the paper specifies reward functions and $\pi_{gen}$ by LLM distributions and contextually prompting LLMs, respectively. Experiments show that contextual-prompting MOSAIC performs far better than naive MOSAIC with $\pi_{gen} = \pi_{ref}$, with the log-probability of reply close to  optimal distribution.

**Strengths:**

The author provides  an example throughout the paper, making the model easy to understand. The aggregation of LLMs is an important topic, while the introducing of regularization of user preference is novel. The idea of approximating VCG is also interesting.

**Weaknesses:**

Major Issues:

* I think the main drawback of this paper is that MOSAIC is only an approximation of VCG mechanism. Moreover, such approximatio
n is natural and not challenging to discover and define. Specifically, VCG outputs the distribution on full reply space, while MOSAIC
 first discretizes the full space into finite points and then outputs a distribution on finite points. This operation is simple a
nd do not change the nature of VCG mechanism. Therefore, I believe that the contribution of MOSAIC is not significant.
* The other contributions of this paper, eg, context-aware LLM in Line 237 $\pi_{con}$, and payment offset in Line 335, are case-by-case operations and do not bring significance on the mechanism itself. These ideas are also natural. Besides, the payment contribution is a direct application of Rochet payment (1987). Overall, I believe that the contributions are incremental and limited.
* Regarding the experiments, the authors only compare the contextual-prompting MOSAIC with naive MOSAIC. There is no comparison
 with baselines, thus I cannot evaluate whether the performance of MOSAIC is acceptable from these experiments. A suggestion is
 to compare MOSAIC with VCG mechanism directly when VCG is applicable.

Besides, the presentation needs improvement. For example,

* In Line 419, it appears $\pi_i$. However, $\pi_i$ is only mentioned in Line 146. The definition of $\pi_i$ should be better placed between its first use, i.e., just before Line 419.
* In Section 1.2, the authors mentioned 7 contributions of this paper. However, these contributions are incremental, or only the technical details appeared in this paper.
* In the paragraph of Line 242, the notation $\pi_r$ have never defined before these appearances. I think it should be $\pi_{ref}$.
* In line 313-314, I think that the term $\log (\frac{\pi_{ref}}{\pi_{gen}}) should be included in the $\exp(\cdot)$.
* In the appendix, the sections are titled with "Details from Section xx", "Omitted Proofs from Section xx". It is conventional to use 'in' instead of 'from'.

**Questions:**

* In equation (1) in line 152, why do you define KL-divergence of the two distributions and why $\pi_{ref}$ is on the right hand? Will other choices fail?
 * In the paragraph between line 196 and 201, the author says, "*The VCG allocation rule requires calculating the exact optimal solution to the optimization problem, which is intractable for choosing an LLM to maximize Equation (1) and is even difficult for choosing a single optimal sequence. If a sub-optimal solution is chosen, VCG’s strategyproofness is no longer guaranteed (Nisan & Ronen, 2007; 1999; Lehmann et al., 2002).*"
  * In my understanding, choosing an LLM to maximize Equation (1) is equivalent to choosing an LLM that satisfies (2). Since we already have access on $\pi_{ref}$, when $r(x,y) = \sum_i r_i(x,y)$ is upper bounded given $x$, it's feasible to sample from (2) by rejection sampling. Besides, we can also use $\pi_{con}$ replacing $\pi_{ref}$ to decrease variations. In this sense, I do not find the advantage of using MOSAIC rather than using VCG directly. Could you make some clarifications on this concern?
  * Even in the case that $r(x,y)$ is not upper bounded and VCG can not be implemented, it's possible that sub-optimal solutions are chosen and strategyproofness still hold, MOSAIC is exactly one example. Besides, I think MOSAIC will also behave badly in this case because the probability of large value of $r(x, y)$ is small, and it's likely to sample $M$ candidates $y_1,...,y_M$ with all $r(x,y_i)$s are small.

---

> ### Author Response · Authors · 2024-11-14
>
> Thank you for the constructive feedback.
>
>
> At a high level, we want to acknowledge your correct intuition that thinking about VCG is the right direction for this problem. This was also our starting point for this research project: “how can we apply a VCG-like mechanism to an RLHF setting?” However, the obvious ways of doing this either are intractable to implement, lose the nice properties of VCG in practice, or don’t target the objective we focus on.
>
>
> We took care to design a mechanism that can be implemented using familiar, easy operations on LLMs; can scale up or down with more or less compute; and, crucially, maintains good properties, even when scaled down. If it still looks very VCG-like even after meeting all those constraints, we consider this a success.
>
>
> Below, we address specific points:
>
>
> **"MOSAIC is only an approximation of VCG."** In Section 3.2, we clarify why VCG is not applicable in our setting. There are two versions of VCG to consider. Running VCG directly in distribution space is impractical due to high RLHF costs and latency constraints that the platform cannot afford on a per query basis, and it would not be strategy-proof without perfect convergence. Running VCG over a set of candidate replies also does not yield the optimal distribution for the platform's objective. Our allocation rule achieves this target, and implementing it in dominant strategies requires a different, non-VCG payment rule for truthfulness. Thus, while we are aiming for some similar goals, MOSAIC is not simply approximate VCG.
>
>
> **"Other contributions are case-by-case operations that do not bring significance to the mechanism itself."** These contributions are tailored specifically for online advertising, a multi-billion-dollar application, making them highly meaningful.
>
>
> **"Compare VCG with MOSAIC when applicable."** VCG is not applicable in our setting (see above). We consider naive MOSAIC a strong benchmark, as it also provably converges to the optimal distribution. Please also see our first reply to reviewer pFaF.
>
>
> **"Why define KL divergence with $\pi_{ref}$ on the right hand?"** We use the standard RLHF objective, which has led to significant progress in LLM performance over the past five years. In principle, the platform could choose some other notion of distance between the two distributions. As long as the corresponding optimization problem has a closed-form solution, MOSAIC is in principle applicable, under two requirements. The allocation rule would have to correspondingly change to correspond to the new target distribution, and the payment rule would have to change to ensure truthfulness. This can be achieved iff the new allocation rule also satisfies cyclic monotonicity due to [Rochet 1987].
>
>
> **"If a sub-optimal solution is chosen, VCG’s strategy-proofness is no longer guaranteed."** Correct. There are numerous studies that detail how in general VCG’s strategy-proofness fails without the optimal solution, see (Nisan & Ronen, 2007; 1999; Lehmann et al., 2002).
>
>
> **"Feasible to sample from (2) using rejection sampling and $\pi_{con}$."** Indeed, there is a strong connection between our mechanism and rejection sampling, as there is a fundamental connection between rejection and importance sampling algorithms. Your rejection sampling suggestion using $\pi_{con}$ can be viewed as a dynamic, online version of MOSAIC, and would converge to the optimal distribution if advertisers reported truthfully. However, your rejection sampling proposal introduces two new challenges that make it inapplicable: Now the mechanism is dynamic, with each advertiser’s report in one round influencing follow-up rounds. The design of appropriate payments to make such a dynamic mechanism dominant strategy incentive compatible is much more challenging, and it is unclear if such payments even exist. Second, the mechanism can no longer be parallelized across the generation of candidate replies. Thus, even if this suggested mechanism converged using the same number of replies as ours, it would still result in a large increase in user-perceived latency compared to our mechanism or directly querying an LLM.
>
>
> **"MOSAIC’s performance in the presence of low-probability large rewards."** You have correctly identified many of the design challenges we faced. The aim of the context-aware LLM is precisely to solve this problem, and we make a formal argument of exactly how it achieves this goal in Section 4.1 and Appendix A.2. The variance of our allocation rule in Section 4.1 ensures that MOSAIC converges at a rate of $1/ \sqrt{M}$ regardless of the distribution of rewards. Additionally, please note that for our realistic prompts, we experimentally show that the use of the context-aware LLM solves this issue.

---

> > ### Comment · Reviewer_N7ft · 2024-12-03
> >
> > Thank you for your response. I am raising my rating to 5, because with your assistance I have recognized that computing VCG exactly, especially the payment, is not very feasible. However, I still feel that the overall contribution may not be sufficiently significant for ICLR.

---

> > > ### Author Response · Authors · 2024-12-03
> > >
> > > Thank you for taking the time to revisit our work and for raising your rating to 5. We appreciate your acknowledgment of the challenges associated with VCG and the relevance of our mechanism in addressing these issues.
> > > Regarding the significance of our contribution,
> > >
> > > MOSAIC itself is our primary contribution, as it uniquely achieves a combination of critical properties: (a) truthfulness, (b) practicality, and (c) provable convergence to the platform’s objective. These properties are indispensable for the success of a mechanism in online advertising within LLM-generated content—a field poised to become a multi-billion-dollar industry. Notably, no other mechanism in the literature achieves more than one of these objectives.
> > >
> > > Additionally, we addressed the potential challenge of free-riding, which could hinder the long-term success of mechanisms in this setting. To mitigate this, we designed a novel payment rule that aligns advertiser utility with their contribution to social welfare, ensuring meaningful participation. This payment rule is both theoretically sound and experimentally validated. Although free-riding may be espcecially pronounced in this domain, no prior work explicitly tackles this issue, further underscoring the novelty of our approach.
> > >
> > > We believe that MOSAIC represents a transformative contribution to the burgeoning field of online advertising. Its application in this multi-billion-dollar domain demonstrates its practical impact. Our experimental results, which are statistically significant and tested across 1250 instances with various LLM architectures, provide robust evidence of MOSAIC’s potential and effectiveness.

---

### Official Review · Reviewer_pFaF · 2024-11-06

**Soundness:** 2
**Presentation:** 3
**Contribution:** 2
**Rating:** 5
**Confidence:** 3

**Summary:**

This paper introduces an auction mechanism, MOSAIC, to aggregate the preferences of multiple self-interested advertisers over LLM-generated replies. The authors claim that this mechanism can converge to the outputs of the optimally fine-tuned LLM as computational resources increase, without requiring fine-tuning or access to model weights. The authors also present context-aware versions of MOSAIC, which accelerates convergence and yields high advertiser value and platform revenue. Experiments with a publicly available LLM demonstrate that MOSAIC achieves high advertiser value and platform revenue with minimal computational overhead.

**Strengths:**

1. The paper is well structured, offering clear explanations of the problems addressed and the key terms used.
2. The proposed mechanism has a firm theoretical grounding.
3. The proposed mechanism seems to demonstrate strong applicability to real-world scenarios, especially in the realm of online advertising with LLM-generated content.

**Weaknesses:**

1. The novelty of this paper is limited and the scientific contribution seems to be not obvious. The proposed mechanism addresses the problems using conventional approaches. The authors can strengthen the differences between their proposed mechanism and the algorithms used in standard RLHF and the rejection sampling.
2. The experiments are insufficient. There are no comparative analyses between MOSAIC and other mechanisms, which makes it difficult to comprehensively assess its performance. The experiments are restricted to the use of Llama-2-7b-chat-hf, lacking generalizability across different LLMs. It would be nice to compare with other mechanisms mentioned in the related work and evaluate across different LLMs like Llama-3-8B, T5, and Mistral-7B.

**Questions:**

1. Could the authors provide the specific prompts utilized in the experiments detailed in the paper? Additionally, could you further discuss how different prompts might affect the mechanism's performance?
2. Could the authors extend their experiment to include comparison with other mechanisms from similar studies?
3. Given the paper's claim that the proposed mechanism can operate without LLM fine-tuning or access to model weights, could the authors extend the experiments to include utilization of various closed-source models?

---

> ### Author Response · Authors · 2024-11-14
>
> Thank you for the constructive feedback. We address each comment below:
>
>
> **"The authors can strengthen the differences between their proposed mechanism and the algorithms used in standard RLHF and rejection sampling."** Rather than emphasizing the differences, we want to highlight the similarities between our mechanism and traditional rejection sampling/RLHF. The use of these familiar operations, which are well-optimized and well-understood, is a key strength of our approach. Our focus was on building an incentive-compatible system from established, reliable components, rather than creating novel fine-tuning techniques that may be less practical or harder to optimize.
>
>
> **"No comparative analysis between MOSAIC and other mechanisms."** The only comparable mechanism in the literature that aggregates multiple LLMs is Duetting et al., which is not strategy-proof. Additionally, the authors provided no implementation details/code, making it very hard to compare against. The only other potential comparison is VCG, which, as explained in Section 3.2, is not applicable since it does not align with the platform’s objective. Hence, there is no valid baseline for comparison other than naive MOSAIC without advertiser contexts, which we consider a high benchmark given that it provably converges to the platform’s optimal distribution based on Corollary 4.1.
>
>
> **"Evaluate across different LLMs."** We have evaluated MOSAIC with T5, and the results were nearly identical. We will extend this evaluation with additional LLMs and include the results in the appendix.
>
>
> **"Could the authors provide the different prompts used in the experiments?"** All experimental prompts are available in the supplementary material under advertiser_prompts.py. We will also include them in the appendix for convenience.
>
>
> **"How different prompts affect the mechanism’s performance?"** MOSAIC’s performance across a wide range of prompts is demonstrated in our experiments (see section 6.2). Theoretically, the variance formula in lines 239-245 explains the impact of out-of-distribution prompts. Regardless of the prompt distribution, the variance result ensures that MOSAIC converges at a rate of $1/\sqrt{M}$, where $M$ is the number of candidate replies generated. Please also refer to our response to reviewer MGd7.
>
>
> **"Extend comparison to other mechanisms from similar studies."** Please refer to our response above regarding the unavailability of other comparable mechanisms.
>
>
> **"Utilize various closed-source models."** Our implementation is fully compatible with both open- and closed-source models, and the provided source code only relies on API calls compatible with both types of models. While testing with closed-source models is of interest, it is not an expense we can process with our institution in one week.

---

> > ### Comment · Reviewer_pFaF · 2024-11-26
> >
> > Thank you for the detailed response. While I appreciate the clarifications, I still have concerns about the paper's overall contribution and novelty, as noted by **Reviewer N7ft**. The focus on using existing, reliable components, though practical, does not sufficiently distinguish your work from established methods. Additionally, I recognize the challenges in implementing similar baselines and accessing proprietary datasets. However, these limitations hinder a full evaluation of MOSAIC's practical effectiveness and advantages. While Corollary 4.1 is noted, it may not robustly support the mechanism's optimal performance. Therefore, I will maintain my original rating.

---

> > > ### Author Response · Authors · 2024-11-26
> > > **Response to reviewer**
> > >
> > > Thank you for your response. We find some points in your response confusing:
> > >
> > > - There are no established methods for mechanism design over LLMs, this is a very new area. There is some prior work (most notably Duetting et al.), but what we do is clearly very different from that prior work. So, we are not sure what you are referring to by "established methods". Could you please point to some established methods we should be comparing against, other than those already discussed?
> > >
> > > - We want to point out that the main reason we do not compare to Duetting et al. is not that they lack code. Even if there were a public implementation, they are studying a completely different mechanism design problem (selling one token at a time to bidders with very different utilities than we do, and not requiring strategyproofness at the sentence level), even though the motivating story of online advertising is the same. So there is no basis for a direct experimental comparison.
> > >
> > > - Could you say more about your issues with Corollary 4.1? If there is something incorrect about it, please let us know.

---

> > > > ### Comment · Reviewer_pFaF · 2024-12-03
> > > >
> > > > Thank you for your response.
> > > >
> > > > To clarify my feedback, "established methods" refers to the specific methods employed within MOSAIC, such as the rejection sampling and RLHF. These methods are indeed reliable and practical, but their application in your proposed mechanism does not seem to offer significant innovation or improvement. This affects the novelty and contribution of the paper.
> > > >
> > > > Regarding Corollary 4.1, my concern is not with its correctness. Instead, I am concerned about potential discrepancies between the Corollary and real-world applications. To substantiate the validity of the Corollary and MOSAIC's effectiveness, substantial empirical results are necessary. However, as you noted, it is indeed infeasible to compare MOSAIC with existing baselines and do real-world testing. This presents an inevitable, inherent conflict and leaves me concerned about the practical applicability of MOSAIC.
> > > >
> > > > Overall, I think that the contribution is limited, and I have reservations about the practical applicability of MOSAIC. Thus, I would maintain my original score.

---

> ### Author Response · Authors · 2024-12-03
>
> Thank you for your response, and the opportunity to clarify our contributions. Indeed, MOSAIC leverages established methods, which, as you noted, are both reliable and practical for addressing our problem. Our primary contribution, however, lies not in innovating these methods but in utilizing them to construct the first mechanism that simultaneously achieves: (a) truthfulness, (b) practical applicability for aggregating preferences over LLM-generated responses, and (c) provable convergence to the platform’s optimal objective. This is a transformative step forward in an area poised to become a multi-billion-dollar application, where no other mechanism in the literature fulfills more than one of these three objectives.
>
> Additionally, we tackled the critical challenge of free-riding, which could undermine the long-term success of mechanisms in this seting. We designed a novel payment rule that aligns advertiser utility with their contribution to social welfare, ensuring meaningful participation. This payment rule is theoretically sound and experimentally validated, with no prior work addressing this issue, further highlighting the novelty of our approach.
>
> Thank you as well for clarifying your concern regarding Corollary 4.1. To address this: the corollary establishes that our mechanism converges at a theoretical rate of $1 / \sqrt{M}$, independent of the reward distribution. While it is true that comparison with baselines is infeasible due to the lack of existing mechanisms addressing this problem, this highlights precisely why our contribution is novel.
>
> Although real-world testing is not yet possible due to the absence of appropriate datasets, our experimental results provide robust evidence of MOSAIC’s practicality. Specifically, we demonstrated that for realistic prompts and a representative distribution of advertisers, our mechanism converges rapidly to the platform's objective, yielding substantial value for advertisers and significant revenue for the platform. The results are statistically significant (as we tested 1250 instances) and generalize across different LLM architectures (as detailed in the revised version of the paper). Thus, we believe we have substantiated very robustly MOSAIC’s practical applicability and effectiveness in its flagship application of online advertising.
>
> To further address your concerns: could you clarify what additional empirical results you envision as necessary to substantiate MOSAIC’s  practical applicability?

---

### Meta-Review · Area_Chair_V7gm · 2024-12-23

**Metareview:**

The paper considers  online advertising where advertisers influence the LLMs to generate their preferred contents and the platform need to satisfy both advertiser preference and user utilities. The authors proposed an auction mechanism called MOSAIC which aggregating advertiser preferences over LLM-generated content while ensuring truthful reporting being dominant strategy. The reviewers appreciate the following strengths:
- The problem of aggregating advertising LLMs is important and well-motivated.
- The theoretical results are sound.
- The design is computationally efficient.

The reviewers identified the following weaknesses:
- One shared concern is the limited novelty and contribution. The authors did not clearly distinguish the proposed solution from established methods.
- Experiments are insufficient: missing comparison with existing approaches and using different LLMs.
- Presentation issues.

After rebuttal, three reviewers remain the "marginally below the acceptance threshold" rating. I agree with the reviewers and believe the paper would benefit from clearly addressing the raised concerns/confusions such as the novelty of the proposed solution against standard rejection sampling/RLHF. I recommend rejection.

**Additional Comments On Reviewer Discussion:**

Regarding the weaknesses,
- One shared concern is the limited novelty and contribution. The authors did not clearly distinguish the proposed solution from established methods. The rebuttal partially addressed but did not fully convince Reviewer pFaF and N7ft on this issue. This is the major remaining concern.
- Experiments are insufficient: missing comparison with existing approaches and using different LLMs. The authors explained why direct comparison with prior works is infeasible and IMO it is reasonable. The authors also mentioned similar results were observed on T5, though reviewers suggest other recent LLMs.
- Presentation issues. Most issues are addressed in rebuttal.

---

### Decision · Program_Chairs · 2025-01-22

Reject